# MULTI-USER REINFORCEMENT LEARNING WITH LOW RANK REWARDS

## ABSTRACT

We consider the problem of collaborative multi-user reinforcement learning. In this setting there are multiple users with the same state-action space and transition probabilities but with different rewards. Under the assumption that the reward matrix of the $N$ users has a low-rank structure – a standard and practically successful assumption in the offline collaborative filtering setting – we design algorithms with significantly lower sample complexity compared to the ones that learn the MDP individually for each user. Our main contribution is an algorithm which explores rewards collaboratively with $N$ user-specific MDPs and can learn rewards efficiently in two key settings: tabular MDPs and linear MDPs. When $N$ is large and the rank is constant, the sample complexity per MDP depends logarithmically over the size of the state-space, which represents an exponential reduction (in the state-space size) when compared to the standard "non-collaborative" algorithms.

## 1 INTRODUCTION

Reinforcement learning has recently seen tremendous empirical and theoretical success Mnih et al. (2015); Sutton et al. (1992); Jin et al. (2020b); Gheshlaghi Azar et al. (2013); Dann & Brunskill (2015). Near optimal algorithms have been proposed to explore and learn a given MDP with sample access to trajectories. In this work, we consider the problem of learning the optimal policies for multiple MDPs collaboratively so that the total number of trajectories sampled per MDP is smaller than the number of trajectories required to learn them individually. This combines reinforcement learning and collaborative filtering. We assume that the various users have the same transition matrices, but different rewards and the rewards have a low rank structure. Low rank assumption is popular in the collaborative filtering literature and has been deployed successfully in a variety of tasks Bell & Koren (2007); Gleich & Lim (2011); Hsieh et al. (2012). This can be regarded as an instance of multi-task reinforcement, various versions of which have been considered in the literature Brunskill & Li (2013); D'Eramo et al. (2020); Teh et al. (2017); Hessel et al. (2019).

**Motivation** Recently, collaborative filtering has been studied in the online learning setting: Bresler & Karzand (2021); Jain & Pal (2022). Here multiple bandit instances are simultaneously explored under low rank assumptions. In this work, we extend this setting to consider stateful modeling of such systems. In the context of e-commerce, this can allow the algorithm to discover temporal patterns like 'User bought a Phone and hence they might be eventually interested in phone cover' or 'User last bought shoes many years ago which might be worn out by now, therefore recommend shoes'. Note that the fact that a user has bought a shoe changes their preferences (and hence the reward function). Our setting allows one to model such changes. While we assume that the users share the same transition matrix, this can be relaxed in practice by clustering users based on side information and modeling each cluster to have a common transition matrix. This approach has been successfully deployed in various multi-agent RL problems in practice, including in sensitive healthcare settings (see Mate et al. (2022) and reference therein).

**Our Contributions** a) *Improved Sample Complexity:* We introduce the setting of multi-user collaborative reinforcement learning in the case of tabular and linear MDPs and provide sample efficient algorithms for both these scenarios without access to a generative model. Under low rank assumption, the total sample complexity required to learn the near-optimal policies for every user scales as $\tilde{O}(N + |\mathcal{S}||\mathcal{A}|)$ instead of $O(N|\mathcal{S}||\mathcal{A}|)$ in the case of tabular MDPs and $\tilde{O}(N + d)$ instead of $O(Nd^2)$ in the case of linear MDPs.

b) *Collaborative Exploration:* In order to learn the rewards of all the users efficiently under low-rank assumptions, we need to deploy standard low rank matrix estimation/completion algorithms, which require specific kinds of linear measurements (See Section 1.1). Without access to a generative model, the main challenge in this setting is to obtain these linear measurements by querying trajectories of carefully designed policies. We design such algorithms in Section 3.

c) *Functional Reward Maximization:* In the case of linear MDPs, matrix completion is more challenging since we observe measurements of the form $e_i^\mathsf{T}\Theta^*\psi$ where $\Theta^* \in \mathbb{R}^{N \times d}$, corresponding to the reward obtained by user $i$, with respect to an embedding $\psi$. Estimating $\Theta^*$ under low rank assumptions requires the distribution of $\psi$ to have certain isotropy properties (see Section 6). Querying such measurements goes beyond the usual reward maximization and are related to mean-field limits of multi-agent reinforcement learning similar to the setting in Cammardella et al. (2020) where a functional of the distribution of the states is optimized. We design a procedure which can sample-efficiently estimate policies which lead to these isotropic measurements (Section 5).

d) *Matrix Completion With Row-Wise Linear Measurements:* For the linear MDP setting, the low rank matrix estimation problem lies somewhere in between the matrix completion (Recht, 2011; Jain et al., 2013) and matrix estimation with restricted strong convexity (Negahban et al., 2009). We give a novel active learning based algorithm where we estimate $\Theta^*$ row by row without any assumptions like incoherence. This is described in Section 6.

## 1.1 RELATED WORKS

**Related Settings:** Multi-task Reinforcement learning has been studied empirically and theoretically Brunskill & Li (2013); Taylor & Stone (2009); D'Eramo et al. (2020); Teh et al. (2017); Hessel et al. (2019); Sodhani et al. (2021). Modi et al. (2017) considers learning a sequence of MDPs with side information, where the parameters of the MDP varies smoothly with the context. Shah et al. (2020) introduced the setting where the optimal Q function $Q^*(s, a)$, when represented as a $\mathcal{S} \times \mathcal{A}$ matrix has low rank. With a generative model, they obtain algorithms which makes use of this structure to obtain a smaller sample complexity whenever the discount factor is bounded by a constant. Sam et al. (2022) improves the results in this setting by considering additional assumptions like low rank transition kernels. Our setting is different in that we consider multiple users, but do not consider access to a generative model. In fact our main contribution is to efficiently obtain measurements conducive to matrix completion without a generative model. Hu et al. (2021) considers a multi-task RL problem with linear function approximation similar to our setting, but with the assumption of low-rank Bellman closure, where the application of the Bellman operator retains the low rank structure. They obtain a bound depending on the quantity $N\sqrt{d}$ instead of $(N + d)$ like in our work. Lei & Li (2019) considers multi-user RL with low rank assumptions in an experimental context.

**Low Rank Matrix Estimation:** Low rank matrix estimation has been extensively studied in the statistics and ML community for decades in the context of supervised learning Candès & Tao (2010); Negahban & Wainwright (2011); Fazel (2002); Chen et al. (2019); Jain et al. (2013; 2017); Recht (2011); Chen et al. (2020); Chi et al. (2019) in multi-user collaborative filtering settings. The basic question is to estimate a $d_1 \times d_2$ matrix $M$ given linear measurements $(x_i^\mathsf{T} M y_i)_{i=1}^n$ when the number of samples is much smaller than $d_1 \times d_2$ using the assumption that $M$ has low rank.

a) *Matrix Completion:* $x_i$ and $y_i$ are standard basis vectors. Typically $x_i$ and $y_i$ are picked uniformly at random and recovery guarantees are given whenever the matrix $M$ is incoherent (Recht, 2011).

b) *Matrix Estimation:* $x_i$ and $y_i$ are not restricted to be standard basis vectors. Typically, they are chosen i.i.d such that the restricted strong convexity holds (Negahban et al., 2009).

In this work, we consider MDPs associated with $N$ users such that their reward matrix satisfies a low rank structure. For the case of tabular MDPs, we use the matrix completion setting and for the case of linear MDPs, our setting lies some where in between settings a) and b) as explained above.

## 1.2 NOTATION

By $\|\cdot\|$ we denote the Euclidean norm and by $e_1, \ldots, e_i, \ldots$ the standard basis vectors of the space $\mathbb{R}^m$ for some $m \in \mathbb{N}$. Let $\mathcal{S}^{d-1} := \{x \in \mathbb{R}^d : \|x\| = 1\}$. Let $\mathcal{B}_d(r) := \{x \in \mathbb{R}^d : \|x\| \leq r\}$. For any $m \times n$ matrix $A$ and a set $\Omega \subseteq [n]$ by $A^\Omega$, we denote the sub-matrix of $A$ where the columns

corresponding to $\Omega^{\complement}$ are deleted. By $\Delta(\mathcal{A})$, we denote the set of all Borel probability measures over the set $\mathcal{A}$. In the sequel,

## 2 PROBLEM SETTING

We consider $N$ users indexed by $[N]$, each of them associated with a horizon $H$ MDP with the same state-space $\mathcal{S}$, the same action space $\mathcal{A}$ and same transition matrices $\mathcal{P} = (P_1, \ldots, P_{H-1})$, where $P_h(\cdot | s_h, a_h)$ is a probability measure over $\mathcal{S}$, which gives the distribution of the state at time $h + 1$ given the action $a_h$ was taken in state $s_h$ at time $h$. Each user has a different reward denoted by $\mathcal{R}_u = (R_{1u}, \ldots, R_{Hu})$ where $R_{hu} : \mathcal{S} \times \mathcal{A} \to [0, 1]$. Denote the MDP associated with the user $u$ by $\mathcal{M}_u := (\mathcal{S}, \mathcal{A}, \mathcal{P}, \mathcal{R}_u)$. For the sake of simplicity, we will assume that the rewards are deterministic.

We assume that all the MDPs start at a random state $S_1$ with the same distribution. Consider a policy $\Pi := (\pi_1, \ldots, \pi_H)$ where $\pi_h : \Delta(\mathcal{A}) \times \mathcal{S} \to \mathbb{R}^+$ is a kernel - i.e, $\pi_h(\cdot | s)$ gives the probability distribution over actions given a state $s$ at time $h$. By $(S_{1:H}, A_{1:H})$ we denote the trajectory $(S_1, A_1), (S_2, A_2), \ldots, (S_H, A_H) \in \mathcal{S} \times \mathcal{A}$. By $(S_{1:H}, A_{1:H}) \sim \mathcal{M}(\Pi)$ we mean the random trajectory under the policy $\Pi$ - where $A_h \sim \pi_h(\cdot | S_h)$ and $S_{h+1} \sim P_h(\cdot | S_h, A_h)$. That is, it is the trajectory of the MDP under the policy $\Pi$. Define the value function of $\mathcal{M}_u$ under policy $\Pi$ as: $V(\Pi, \mathcal{M}_u) := \mathbb{E}_{(S_{1:H}, A_{1:H}) \sim \Pi} \sum_{h=1}^{H} R_{hu}(S_h, A_h)$. We will call a policy $\hat{\Pi}_u$ to be $\epsilon$ optimal for $\mathcal{M}_u$ if $V(\hat{\Pi}_u, \mathcal{M}_u) \geq \sup_\Pi V(\Pi, \mathcal{M}_u) - \epsilon$. Our goal is to find $\epsilon$ optimal policies for every $u \in [N]$ with as few samples as possible under low rank assumptions on the rewards $R_{uh}$.

**Reward Free Exploration:** The objective of reward free RL is to explore an MDP such that we can obtain the optimal policy for every possible reward. After collecting $K$ trajectories from the MDP sequentially (denoted by $\mathcal{D}_K$), the algorithm outputs functions $\hat{\Pi}$ and $\hat{V}$ whose input is a reward function $\mathcal{R} = (R_h(\cdot, \cdot))_{h=1}^H$ (bounded between $[0, 1]$) and the output is a nearly-optimal policy $\hat{\Pi}(\mathcal{R})$ and its estimated value $\hat{V}(\hat{\Pi}(\mathcal{R}))$ for this reward function. Denote the MDP with this reward function by $\mathcal{M}_\mathcal{R}$. Given $\epsilon > 0$ and $\delta \in [0, 1]$, we let $K^{\mathsf{rf}}(\epsilon, \delta)$ to be such that whenever $K \geq K^{\mathsf{rf}}(\epsilon, \delta)$, with probability at-least $1 - \delta$ we have:

a) $\sup_\mathcal{R} |V(\hat{\Pi}, \mathcal{M}_\mathcal{R}) - \hat{V}(\hat{\Pi}(\mathcal{R}))| \leq \epsilon$ and b) $\hat{\Pi}$ is an $\epsilon$ optimal policy for $\mathcal{M}_\mathcal{R}$ for every $\mathcal{R}$.

This setting was introduced in Jin et al. (2020a). In this work, we will use the reward free exploration algorithms in Zhang et al. (2020) for tabular MDPs and Wagenmaker et al. (2022) for linear MDPs.

**Tabular MDP Setting** $\mathcal{S}$ and $\mathcal{A}$ are finite sets. Denote the reward $R_{hu}(s, a)$ by the $N \times |\mathcal{S}||\mathcal{A}|$ matrix $R_h$ where $R_h(u, (s, a)) = R_{hu}(s, a)$. We have the following low-rank assumption:

**Assumption (Tab) 1.** *The matrix $R_h$ has rank $r$ for some $r \leq \frac{1}{2} \min(N, |\mathcal{S}||\mathcal{A}|)$.*

**Linear MDP Setting** Our definition is slightly different from the one in Jin et al. (2020b). In this setting, we consider embeddings $\phi : \mathcal{S} \times \mathcal{A} \to \mathbb{R}^d$, $\psi : \mathcal{S} \times \mathcal{A} \to \mathbb{R}^d$ such that $\|\phi(s, a)\|_1 \leq 1$, $\|\psi(s, a)\|_2 \leq 1$. We make the following assumptions:

1. There exists $\theta_{hu} \in \mathbb{R}^d$, $\|\theta_{hu}\|_2 \leq \sqrt{d}$ such that $R_{hu}(s, a) = \langle \theta_{hu}, \psi(s, a) \rangle$ and $R_h(s, a) \in [0, 1]$.

2. There exist signed measures $\mu_{1h}, \ldots, \mu_{dh}$ over the space $\mathcal{S}$ such that: $P_h(\cdot | s, a) = \sum_{i=1}^d \mu_{ih}(\cdot) \langle \phi(s, a), e_i \rangle$

We will assume that $\mu_i$ are such that $\| \int \mu_{ih}(ds) \phi(s, a) \pi(da|s) \|_1 \leq 1$ and $\sup_{i,h} \int |\mu_{ih}(ds)| \leq C_\mu$. This is true whenever $\mu_{ih}$ are probability measures. We consider different embeddings for transition ($\phi$) and reward ($\psi$) as the transition embeddings have a natural $\| \cdot \|_1$ structure since they give linear combinations of measure which make up $P_h(\cdot | s, a)$. We denote the $N \times d$ matrix whose $u$-th row is $\theta_{uh}^\mathsf{T}$ to be $\Theta_h$. The low-rank assumption in this setting takes the following form:

**Assumption (Lin) 1.** . *The $N \times d$ matrix $\Theta_h$ has rank $r \leq \frac{1}{2} \min(N, d)$.*

For the task of reward maximization in Linear MDPs, deterministic policies of the form $\pi_h(s) = \arg \sup_a \langle \psi(s, a), u_h^* \rangle + \langle \phi(s, a), v_h^* \rangle$ suffice, for $u_h^*, v_h^* \in \mathbb{R}^d$. We want to complete the matrix $\Theta_h$,

with data of the form $(e_i, \psi(S_h, A_h), e_i^\mathsf{T} \Theta_h \psi(S_h, A_h))$. To achieve matrix estimation, we need to query $(S_h, A_h)$ such that the distribution of $\psi(S_h, A_h)$ is 'nearly isotropic' (See Section 6). Such $(S_h, A_h)$ cannot necessarily be obtained as a result of reward maximization policy of a single agent and is related to mean-field limit of multi-agent RL (similar to the setting in Cammardella et al. (2020)), and it could necessarily be a randomized policy. However, the space of all possible policies can be very large and intractable. Therefore, we restrict our attention to policies given by some fixed policy space $\mathbb{Q}$. With some abuse of notation, we define the total variation distance between two kernels as: $\mathsf{TV}(\pi_h, \pi'_h) := \sup_{s \in \mathcal{S}} \mathsf{TV}(\pi_h(\cdot|s), \pi'_h(\cdot|s))$. We define a distance over $\mathbb{Q}$ by $D_{\mathbb{Q}}(\Pi^{(1)}, \Pi^{(2)}) = \sup_{h \in [H]} \mathsf{TV}(\pi_h^{(1)}, \pi_h^{(2)})$, where $\Pi^{(i)} = (\pi_h^{(i)} : h \in [H])$. We refer to Section A for additional discussion on the above observations, connections to multi-agent mean-field RL and construction of $\mathbb{Q}$ such that it contains all $\epsilon$-optimal policies for every possible linear reward.

# 3 THE ALGORITHM

Our algorithm proceeds in 4 phases. Note that since all the users in our setting have the same transition probabilities, we can run reward free RL (Phase 1) in a distributed fashion over the users. Reward free exploration is useful in this setting since all users share the same MDP and the main unknown is the reward matrix. Reward free exploration can be done collectively by selecting random users instead of the same user, which reduces the per user complexity. This is then used in two ways: 1) Collaboratively exploring the space in order to complete the reward matrix (Phase 2) 2) Learning the optimal reward for every user once the reward matrix is known (Phase 4).

## 3.1 TABULAR MDP CASE:

**Phase 1: Reward Free Exploration** We run the reward free RL algorithm in Zhang et al. (2020) for $K^{\mathsf{rf}}\left(\frac{\epsilon}{8}, \frac{\delta}{2}\right) = C \frac{|\mathcal{S}||\mathcal{A}|H^2\left(|\mathcal{S}| + \log\left(\frac{1}{\delta}\right)\right)}{\epsilon^2} \mathrm{polylog}\left(\frac{|\mathcal{S}||\mathcal{A}|H}{\epsilon}\right)$ time steps by picking the MDP corresponding to a uniformly random user whenever the reward free RL algorithm queries a trajectory. Let the output of the reward free RL algorithm be $\hat{\Pi}$ and $\hat{V}$.

**Phase 2: Querying the Reward Matrix** In this phase we query a 'uniform mask' with the parameter $p$ for the reward matrix $R_h$ using Algorithm 1. For each $(s, a) \in \mathcal{S} \times \mathcal{A}$ and $h \in [H]$, maintain a counter $T_{h,(s,a)}$ for $(s, a) \in \mathcal{S} \times \mathcal{A}$ and $h \in [H]$, initialized at 0. Given the 'active sets' $\mathcal{G} = (\mathcal{G}_h)_{h \in [H]} \subseteq \mathcal{S} \times \mathcal{A}$ and $h \in [H]$, we define the reward $\mathcal{J}(; \mathcal{G}) = (J_1, \dots, J_h)$ by

$$J_h(s, a; \mathcal{G}) = \mathbb{1}((s, a) \in \mathcal{G}_h) \tag{1}$$

We will denote this reward by $\mathcal{J}(\cdot; \mathcal{G})$. Initialize active set $\mathcal{G} = (\mathcal{G}_h)_{h \in [H]}$ such that $\mathcal{G}_h = \mathcal{S} \times \mathcal{A}$. We initialize the reward matrix $\hat{R}_h(u, (s, a)) = *$, where $*$ denotes unknown entry. This algorithm terminates when it detects that sufficient number of samples have been collected for matrix completion.

**Phase 3: Reward Matrix Completion** We receive $\mathcal{G}_h$ and the partially observed matrix $\hat{R}_h$ for each $h$ as the output of Algorithm 1. By $\hat{R}_h^{\mathcal{G}_h^{\complement}}$, we denote the sub-matrix where the columns corresponding to $\mathcal{G}_h$ are deleted. We use the nuclear norm minimization algorithm given in Recht (2011) to recover $R_h^{\mathcal{G}_h^{\complement}}$ from $\hat{R}_h^{\mathcal{G}_h^{\complement}}$ for every $h \in [H]$.

**Phase 4: Computing the Optimal Policy** Phase 3 outputs the completed sub-matrix $R_h^{\mathcal{G}_h^{\complement}}$, where only the columns corresponding to $|\mathcal{G}_h^{\complement}|$ are recovered. We construct the recovered matrix $\bar{R}_h$ by setting $\bar{R}_h^{\mathcal{G}_h^{\complement}} = R_h^{\mathcal{G}_h^{\complement}}$ and $\bar{R}_h^{\mathcal{G}_h} = 0$. We compute the optimal policy for each user using the rewards from $\bar{R}_h$ via the output of the reward free RL, $\hat{\Pi}$, from Phase 1.

## 3.2 LINEAR MDP CASE:

**Phase 1 : Reward Free RL** We run the reward free RL algorithm for Linear MDPs from Wagenmaker et al. (2022), with error $\epsilon$ and probability of failure $\frac{\delta}{4}$. We use trajectories from random users whenever a trajectory is queried. Here, $K^{\mathsf{rf}}(\epsilon, \delta/4) = \frac{CdH^5(d + \log(\frac{1}{\delta}))}{\epsilon^2} + \frac{Cd^{9/2}H^6 \log^4(\frac{1}{\delta})}{\epsilon}$.

**Output:** Active sets $\mathcal{G} = (\mathcal{G}_h)_{h \in [H]}$, Partially complete matrix $\hat{R}_h$

$t \leftarrow 0$ ;

$\hat{P}^{\mathcal{G}} \leftarrow \hat{V}(\mathcal{J}(\cdot; \mathcal{G})), \hat{\Pi}^{\mathcal{G}} \leftarrow \hat{\Pi}(\mathcal{J}(\cdot; \mathcal{G}))$;

**while** $\hat{P}^{\mathcal{G}} > \frac{\epsilon}{2}$ **do**

  $U_t \leftarrow \mathsf{Unif}([N])$ ;        // Pick a user uniformly at random

  $S_{1:H}^{(t)}, A_{1:H}^{(t)}, R_{1:H}^{(t)} \sim \mathcal{M}_{U_t}(\hat{\Pi}^{\mathcal{G}})$ ;        // Query trajectory

  **for** $h \in [H]$ **do**

   **if** $(S_h^{(t)}, A_h^{(t)}) \in \mathcal{G}_h$ *and* $R_h(U_t, (S_h^{(t)}, A_h^{(t)})) = *$ **then**

    $T_{h,(S_h^{(t)}, A_h^{(t)})} \leftarrow T_{h,(S_h^{(t)}, A_h^{(t)})} + 1$ ;      // Increment count

    $\hat{R}_h(U_t, (S_h^{(t)}, A_h^{(t)})) \leftarrow R_h^{(t)}$ ;      // Fill Missing Entry

   **end**

   **if** $T_{h,(S_h^{(t)}, A_h^{(t)})} = Np$ **then**

    $\mathcal{G}_h \leftarrow \mathcal{G}_h \setminus \{(S_h^{(t)}, A_h^{(t)})\}$ ;    // Remove element from Active Set

   **end**

  **end**

  $t \leftarrow t + 1$ ;

  $\hat{P}^{\mathcal{G}} \leftarrow \hat{V}(\mathcal{J}(\cdot; \mathcal{G}))$;

  $\hat{\Pi}^{\mathcal{G}} \leftarrow \hat{\Pi}(\mathcal{J}(\cdot; \mathcal{G}))$;

**end**

**Algorithm 1:** Uniform Mask Sampler for Tabular MDPs

**Phase 2: Querying Linear Measurements of the Reward Matrix** We obtain policies whose trajectory data allows low rank matrix estimation of the reward matrix.

**Step 1:** For each time step $h \in [H]$, we want to query obtain samples $(s_h^{(t)}, a_h^{(t)})$ such that $\sum_{t=1}^{T} \phi(s_h^{(t)}, a_h^{(t)}) \phi^{\mathsf{T}}(s_h^{(t)}, a_h^{(t)}) \succeq \kappa^2 I$. This can be done by Algorithm 2. Given a projector $Q$ to some subspace of $\mathbb{R}^d$, by $\mathcal{Q}_{h,Q}$ denote the reward $\|Q\phi(s,a)\|^2$ at time $h$ and 0 otherwise. The termination condition ensures that we see enough data in all directions $\phi$, which allows us to find collaborative exploration policy below.

**Step 2:** Using the observations given in Step 1, we compute the policy $\hat{\Pi}^{f,h}$ which approximately satisfies the property given in Assumption 3. This procedure is described in Section 5.

**Phase 3: Estimating Low Rank Reward Matrix** For this, we use the active learning procedure given in Section 6 via row-wise linear measurements along with the policy $\Pi^{\mathsf{MC},h} = \hat{\Pi}^{f,h}$, which was computed in Phase 2.

**Phase 4: Computing the Optimal Policy** Once the reward matrix $\Theta_h$ have been reconstructed for every $h$ in Phase 3, we use the output of reward free RL in order to compute the $\epsilon$ optimal policy for each user.

# 4 MAIN RESULTS

## 4.1 TABULAR MDP:

The standard assumption for low rank matrix completion is that of incoherence, which ensures that the matrix is not too sparse so that sparse measurements are sufficient to learn it. The following definition is used in Recht (2011).

**Definition 1.** *Given a $r$ dimensional sub-space $U$ of $\mathbb{R}^n$, we define the coherence of as:*

$$\mu(U) := \frac{n}{r} \sup_{1 \le i \le n} \|P_U e_i\|^2 .$$

*A $n_1 \times n_2$ matrix $M$ with singular value decomposition $U\Sigma V^{\mathsf{T}}$ is called $(\mu_0, \mu_1)$ coherent if:*

**Input:** Total time $T$; Tolerance $\gamma$; lower isometry constant $\kappa$
**Output:** $\phi_{ht}, S_{(h+1)t}$ for $h \in [H-1], t \in [T]$
$h \leftarrow 1$ ;
**while** $h \leq H - 1$ **do**
    $Q \leftarrow I$ ;
    $\hat{\Pi}^Q \leftarrow \hat{\Pi}(\mathcal{Q}_{h,Q})$ ;
    $G_{\phi,h} \leftarrow 0$ ;                                `// Grammian initialized to 0`
    $t \leftarrow 1$ ;
    **while** $t \leq T$ **do**
        $U_t \sim \mathsf{Unif}([N])$ ;                    `// Pick a uniformly random user`
        $S_{1:H}, A_{1:H} \sim \mathcal{M}_{U_t}(\hat{\Pi}^Q)$ ;           `// Obtain Trajectory`
        $\phi_{ht} \leftarrow \phi(S_h, A_h)$ ;
        $S_{(h+1)t} \leftarrow S_{h+1}$ ;
        $G_{\phi,h} \leftarrow G_{\phi,h} + \phi_{ht}\phi_{ht}^{\mathsf{T}}$ ;           `// Update Grammian`
        **if** $\exists y \in \mathbb{S}^{d-1} : y^{\mathsf{T}} G_{\phi,h} y < \kappa^2$ **then**
            $Q \leftarrow$ the eigenspace of $G_{\phi,h}$ with eigenvalues $< \kappa^2 I$;
            $\hat{\Pi}^Q \leftarrow \hat{\Pi}(\mathcal{Q}_{h,Q})$ ;
        **end**
        $t \leftarrow t + 1$
    **end**
    $h \leftarrow h + 1$
**end**

**Algorithm 2:** Well Conditioned Matrix Sampler

*a) The coherence of the row and column spaces of $M$ are at-most $\mu_0$ b) The absolute value of every entry of $UV^{\mathsf{T}}$ is bounded above by $\mu_1 \sqrt{\frac{r}{n_1 n_2}}$.*

Given a policy $\Pi$, and $\Omega \subseteq \mathcal{S} \times \mathcal{A}$, by $P_h^{\Pi}(\Omega)$ we denote the probability that at time $h$ we have $(S_h, A_h) \in \Omega$ under the policy $\Pi$.

**Assumption (Tab) 2.** *Given the reward matrix $R_h$ and $\Omega_h \subset \mathcal{S} \times \mathcal{A}$, recall the notation for the sub-matrix $R_h^{\Omega_h}$ of $R_h$. If $\sup_{\Pi} P^{\Pi}(\Omega_h^{\complement}) < \epsilon$ have:*

1. *$R_h^{\Omega_h}$ is $(\mu_0, \mu_1)$ incoherent.*

2. *$|\Omega_h^{\complement}| \leq \frac{|\mathcal{S}|}{2}$*

The incoherence assumption for $R_h^{\Omega}$ makes sense since the set $\Omega^{\complement}$ cannot be easily reached with *any* policy with a probability larger than $\epsilon$. In fact we can arrive at an $\epsilon$ optimal policy for the original reward by just setting the rewards at $\Omega^{\complement}$ to be $0$. These can be thought of as redundant states which do not matter for our RL model with *any* reward.

**Theorem 1.** *Suppose Assumption (Tab) 1, 2 hold. Let the parameter $p = C \frac{\max(\mu_1^2, \mu_0) r(N + |\mathcal{S}||\mathcal{A}|) \log^2 |\mathcal{S}||\mathcal{A}| \log(\frac{H}{\delta})}{N|\mathcal{S}||\mathcal{A}|}$ . for some large enough constant $C$. Assume that $|\mathcal{S}||\mathcal{A}|$ and $N$ are large enough such that $p < 1/2$. Then, with probability at-least $1 - \delta$, we can find an $\epsilon$ optimal policy $\hat{\Pi}_u$ for every user $u \in [N]$ whenever the total number of trajectories queried is:*

$$C \frac{|\mathcal{S}||\mathcal{A}|H^2 \left(|\mathcal{S}| + \log(\frac{1}{\delta})\right)}{\epsilon^2} \mathsf{polylog}(\frac{|\mathcal{S}||\mathcal{A}|H}{\epsilon}) + \frac{C \max(\mu_1^2, \mu_0) r(N + |\mathcal{S}||\mathcal{A}|) H \log^2 |\mathcal{S}||\mathcal{A}| \log(\frac{H}{\delta})}{\epsilon}$$

**Remark 1.** *For large $N$, the number of trajectories queried per user is $\tilde{O}(\frac{rH \log^2(|\mathcal{S}||\mathcal{A}|)}{\epsilon})$, which is an exponential improvement in the state-space size dependence when compared to the minimax rate of $\frac{|\mathcal{S}||\mathcal{A}|H^2}{\epsilon^2}$ (Dann & Brunskill, 2015). Every phase in the algorithm has polynomial computational complexity in $N, |\mathcal{S}||\mathcal{A}|$ and $\frac{1}{\epsilon}$. The probability $p$ is chosen such that $p|\mathcal{S}||\mathcal{A}|N = \tilde{O}(r(|\mathcal{S}||\mathcal{A}| + N))$, which is the number of free parameters required to describe a rank $r$ matrix.*

## 4.2 LINEAR MDP

**Assumption (Lin) 2.** *There exists a $\gamma > 0$ such that for every $x \in \mathcal{S}^{d-1}$, and every $h \in [H]$ there exists a policy $\Pi_{x,h}$ such that whenever $S_{1:H}, A_{1:H} \sim \Pi_{x,h}$, $\mathbb{E}\langle \phi(S_h, A_h), x\rangle^2 \geq \gamma$*

The assumption above shows that we can obtain information about all directions. If this does not hold for any $\gamma$, then $\phi(S_h, A_h)$ does not have any component in some direction $x_0$ with *any* policy. Thus, we can remove the sub-space spanned by $x_0$ and make the embedding space $\mathbb{R}^{d-1}$ at time $h$.

**Assumption (Lin) 3.** *There exist $\zeta, \xi > 0$ such that for every $h \in [H]$, there exists a policy $\Pi^{h,\zeta,\xi} \in \mathbb{Q}$ such that whenever $S_{1:H}, A_{1:H} \sim \mathcal{M}(\Pi^{h,\zeta,\xi})$, we have:*

$$\inf_{x \in \mathcal{S}^{d-1}} \mathbb{E}\left[|\langle x, \psi(S_h, A_h)\rangle|\sqrt{d} - \xi d\langle x, \psi(S_h, A_h)\rangle^2\right] \geq \zeta$$

The assumption above ensures that there exist measurements $\psi(S_h, A_h)$ which are conducive to low rank matrix estimation as considered in Section 6.

**Assumption (Lin) 4.** *For any $1 \geq \eta > 0$, there exists an $\eta$ net for $\mathbb{Q}$, denoted by $\hat{\mathbb{Q}}_\eta$ such that $\log|\hat{\mathbb{Q}}_\eta| \leq D\log(\frac{1}{\eta})$.*

We refer to Section A, where we justify this assumption. We first demonstrate that deterministic policies which are sufficient for reward maximization (as used in Jin et al. (2020b)) cannot be used in this context, so a set of stochastic policies is required. We then construct such policy classes with $D = O(dH \log dH \log \log(|\mathcal{A}|))$.

**Theorem 2.** *Suppose Assumptions (Lin) 1 2 3 and 4 hold and suppose $\epsilon < \frac{\gamma}{2}$. In Algorithm 2, we set $\kappa = \frac{CC_\mu dH \sqrt{dH+D}(\sqrt{d}+\xi d)}{\zeta}\sqrt{\log\left(\frac{C_\mu H(d+D)}{\zeta\gamma\delta}\right)}$ and $T = C\frac{\kappa^2 d}{\gamma^2}\log\frac{d\kappa}{\gamma}$.*

*Then, with probability at least $1 - \delta$, our algorithm finds $\epsilon$ optimal policy for every user $u \in [N]$ with the total number of trajectories being bounded by: $T_{\mathsf{rf}} + T_{\mathsf{pol}} + T_{\mathsf{mat-comp}}$, where:*

$$T_{\mathsf{rf}} = \frac{dH^5(d+\log(\frac{1}{\delta}))}{\epsilon^2} + \frac{d^{9/2}H^6\log^4(\frac{1}{\delta})}{\epsilon}, \quad T_{\mathsf{pol}} = \frac{C_\mu^2 d^5 H^3(dH+D)\log^2\left(\frac{C_\mu H(d+D)}{\zeta\gamma\delta}\right)}{\zeta^2\gamma^2} \quad T_{\mathsf{mat-comp}} = C\frac{Hr(N+d\log N)\log\frac{d}{\zeta\xi} + H\log N\log\left(\frac{\log N}{\delta}\right)}{\zeta^2\xi^2}$$

**Remark 2.** *Note that whenever $N$ is large, we have the sample complexity per user to be $O(Hr)$. This is much better than the dependence of $\Omega(d^2 H^2)$ in the minimax lower bounds for Linear MDPs (Wagenmaker et al., 2022). While Phases 1 and 2 of the algorithm have a computational complexity which is polynomial in $d$ and $\frac{1}{\epsilon}$, the optimization problems posed in Phase 3 and 4 are not necessarily polynomial time. We leave the computational aspects to future work. Notice that the sample complexity $\tilde{O}(r(N+d))$ corresponds to the number of free parameters required to describe a rank $r$ matrix.*

## 5 OBTAINING POLICIES WITH GIVEN STATISTICS

In this section, we consider the Linear MDP setting and describe the sub-routine described in Step 2 of Phase 2 of the algorithm where we compute a policy $\hat{\Pi}^{f,h}$ such that the law of $\phi(S_h, A_h)$ under this policy approximately satisfies the property given in Assumption 3. This is required in order to use the guarantees for low matrix estimation in Phase 3, which is described in Section 6. We first state a structural lemma which characterizes the law of $S_{h+1}, A_{h+1}$ under any policy $\Pi$.

**Lemma 1.** *Consider any policy $\Pi = (\pi_1, \ldots, \pi_{H-1}, \pi_H)$ to the MDP $\mathcal{M}$. Let $S_{1:H}, A_{1:H} \sim \mathcal{M}(\Pi)$. Then for any bounded, measurable function $g : \mathcal{S} \times \mathcal{A} \to \mathbb{R}$, we have:*

$$\mathbb{E}g(S_{h+1}, A_{h+1}) = \sum_{i=1}^{d} \nu_i \int g(s, a)d\mu_{ih}(ds)\pi_{h+1}(da|s)$$

*Where $\nu_i := \langle \mathbb{E}\phi(S_h, A_h), e_i\rangle$*

We now want to estimate certain statistics under any policy using available data, obtained from the output of Algorithm 2. Notice that the output of Algorithm 2 gives a sequence of random variables $(\phi_{h1}, s_{(h+1)1}), \ldots, (\phi_{hT}, s_{(h+1)T}) \in \mathbb{R}^d \times \mathcal{S}$ such that $(s_{(h+1)l})_{l=1}^{T} | (\phi_{hl})_{l=1}^{T} \sim \prod_{l=1}^{T} \left( \sum_{i=1}^{d} \langle \phi_{hl}, e_i \rangle \mu_{hi}(\cdot) \right)$ and $G_{\phi,h} := \sum_{t=1}^{T} \phi_{ht} \phi_{ht}^{\mathsf{T}}$. For any measurable function $g : \mathcal{S} \times \mathcal{A} \to \mathbb{R}^K$, $\nu \in \mathbb{R}^d$ such that $\|\nu\| \leq 1$ and any randomized policy $\Pi = (\pi_1, \ldots, \pi_H)$ we define:

1. $\mathcal{T}_1(g, \pi_1) = \mathbb{E} \int g(S_1, a) \pi_1(da | S_1)$
2. $\mathcal{T}_{h+1}(g; \nu, \pi_{h+1}) = \sum_{i=1}^{d} \langle \nu, e_i \rangle \int \mu_{ih}(ds) \pi_{h+1}(da|s) g(s, a)$ when $h \leq H - 1$
3. $E_1^\nu(\Pi) := \|\mathcal{T}_1(\phi, \pi_1) - \nu\|_1$
4. $E_h^\nu(\Pi) = \inf_{\nu_1, \ldots, \nu_{h-1} \in \mathcal{B}_d(1)} F(\Pi, \nu_1, \ldots, \nu_{h-1}, \nu)$ whenever $h > 1$

Define $\alpha_{ht,\nu} = \phi_{ht}^{\mathsf{T}} G_{\phi,h}^{-1} \nu$. We estimate these operators from data as follows:

1. $\hat{\mathcal{T}}_1(g, \pi_1) = \frac{1}{T} \sum_{t=1}^{T} \int g(s_{1t}, a) \pi_1(da | s_{1t})$
2. $\hat{\mathcal{T}}_{h+1}(g, \nu, \pi_{h+1}) = \sum_{t=1}^{T} \alpha_{ht,\nu} \int g(s_{(h+1)t}, a) \pi_{h+1}(da | s_{(h+1)t})$
3. $\hat{E}_1^\nu(\Pi) := \|\hat{\mathcal{T}}_1(\phi, \pi_1) - \nu\|_1$
4. $\hat{E}_h^\nu(\Pi) = \inf_{\nu_1, \ldots, \nu_{h-1} \in \mathcal{B}_d(1)} \hat{F}(\Pi, \nu_1, \ldots, \nu_{h-1}, \nu)$ whenever $h > 1$

Where, for $h > 1$ and $\nu_1, \ldots, \nu_{h-1} \in \mathcal{B}_d(1)$, we have defined:

1. $F(\Pi, \nu_1, \ldots, \nu_{h-1}, \nu_h) := E_1^{\nu_1}(\Pi) + \sum_{j=2}^{h} \|\mathcal{T}_j(\phi, \nu_{j-1}, \pi_j) - \nu_j\|_1$
2. $\hat{F}(\Pi, \nu_1, \ldots, \nu_{h-1}, \nu_h) := \hat{E}_1^{\nu_1}(\Pi) + \sum_{j=2}^{h} \|\hat{\mathcal{T}}_j(\phi, \nu_{j-1}, \pi_j) - \nu_j\|_1$

Define $f(s, a; x) := |\langle x, \psi(s, a) \rangle| \sqrt{d} - \xi d \langle x, \psi(s, a) \rangle^2$. The output of our method is:

1. $\hat{\Pi}^{f,1} = \arg\sup_{\Pi = (\pi_1, \ldots, \pi_H) \in \mathbb{Q}} \inf_{x \in \mathcal{S}^{d-1}} \hat{\mathcal{T}}_1(f(\cdot; x), \pi_1)$
2. $(\hat{\Pi}^{f,h}, \hat{\nu}) = \arg\sup_{\substack{\nu \in \mathcal{B}(1) \\ \Pi = (\pi_1, \ldots, \pi_H) \in \mathbb{Q}}} \inf_{x \in \mathcal{S}^{d-1}} \hat{\mathcal{T}}_h(f(\cdot; x); \nu, \pi_h)$ whenever $h > 1$, subject to $\hat{E}^{\hat{\nu}, h-1}(\hat{\Pi}^{f,h}) \leq \eta_0$
3. Assign output: $\hat{\Pi}^{\zeta, \xi, h} = \hat{\Pi}^{f,h}$

The idea behind this method is as follows. First, using the output of algorithm 2, we construct $\hat{\mathcal{T}}_h(g, \nu, \pi_h)$, which approximates the functional $\mathcal{T}_h(g, \nu, \pi_h)$ uniformly for every $\nu, \pi_h$. This is shown in Lemma 9 in the appendix. We will show in Theorem 3 that obtaining policies which can be used with the matrix completion routine reduces to picking a policy $\Pi^{f,h} = (\pi_1, \ldots, \pi_H)$ such that whenever $S_{1:H}, A_{1:H} \sim \mathcal{M}(\Pi^{f,h})$, we must have: $\mathbb{E} \inf_{x \in \mathcal{S}^{d-1}} f(S_h, A_h; x) \geq \zeta$. Now, we use Lemma 1 to conclude that if such a policy exists, then there exist $\nu_1, \ldots, \nu_{h-1}$ such that $\mathbb{E}\phi(S_j, A_j) = \nu_j$ and $\inf_{x \in \mathcal{S}^{d-1}} \mathcal{T}_h(f(; x); \nu_h, \pi_h) \geq \zeta$. Since we only have sample access, we find such a policy approximately by optimizing using the estimates $\hat{\mathcal{T}}$ instead of the exact functional $\mathcal{T}$ as described above.

**Theorem 3.** *We condition on the event $G_{\phi,h} \geq \kappa^2 I$ for every $h \in [H]$. Let $\kappa, \eta, \eta_0$ be such that for some small enough constants $c_0, c > 0$ and a large enough constant $C > 0$:*

*1.* $\eta \leq c \frac{\zeta}{C_\mu dH(\sqrt{d} + \xi d) H} \sqrt{\frac{\kappa^2}{T}}; \quad \eta_0 = c_0 \frac{\zeta}{C_\mu(\sqrt{d} + d\xi)}$

*2.* $\kappa \geq C \frac{C_\mu(\sqrt{d} + \xi d) dH}{\zeta} \sqrt{\log\left(\frac{dH|\hat{\mathbb{Q}}_\eta|}{\delta}\right) + dH \log\left(\frac{d}{\eta}\right)}$

*Recall the policy $\hat{\Pi}^{f,h}$. Suppose the Assumption 3 holds. Then, with probability at-least $1 - \frac{\delta}{4}$ we obtain the policy $\hat{\Pi}^{f,h}$ is such that whenever $S_{1:H}, A_{1:H} \sim \mathcal{M}(\hat{\Pi}^{f,h})$, we have:*

$$\mathbb{E} \inf_{x \in \mathcal{S}^{d-1}} f(S_h, A_h; x) \geq \frac{\zeta}{2}$$

*This implies that $\psi(S_h, A_h)$ satisfies $\mathbb{E}|\langle \psi(S_h, A_h), x \rangle| \geq \frac{\zeta}{2\sqrt{d}}; \quad \mathbb{E}\psi(S_h, A_h)\psi_{ik}^\intercal \leq \frac{1}{d\xi^2}$*

## 6 Matrix Estimation with Row-wise Linear Measurements

In this section, we describe the active learning based low rank matrix estimation procedure. For an unknown rank $r$ matrix $\Theta^*$ (corresponding to the reward matrix $\Theta_h^*$ in the definition of Linear MDPs) of dimensions $N \times d$, we are allowed to query samples of the form $(e_i, \psi, e_i^\intercal \Theta^* \psi)$ for any $i \in [N]$ of our choice and $\psi = \psi(S_h, A_h)$ where $S_{1:H}, A_{1:H} \sim \mathcal{M}(\Pi^{\mathsf{MC},h})$, for some input policy $\Pi^{\mathsf{MC},h}$. This corresponds to running the MDP corresponding to user $i$, with the policy $\Pi^{\mathsf{MC},h}$ and observing the reward at time $h$, given by $\langle e_i, \Theta_h^* \Psi(S_h, A_h) \rangle$. Our basic task is to estimate the matrix $\Theta^*$ from these samples with high-probability.

### 6.1 The Estimator

Given any $N \times d$ matrix $\Delta$, by $\Delta_i^\intercal$, we denote its $i$-th row. Given $K \in \mathbb{N}$, and a sequence of vectors $\Psi = (\psi_{ik} \in \mathbb{R}^d)_{i \in [N], k \in [K]}$.

$$L(\Delta, \Psi) := \frac{1}{NK} \sum_{i=1}^{N} \sum_{k=1}^{K} |\langle \Delta_i, \psi_{ik} \rangle|^2$$

We estimate $\Theta^*$ row-wise using the following iterative procedure, where recover some rows of $\Theta^*$ into $\hat{\Theta}$ in each iteration and obtain the corresponding linear measurements of $\Theta^*$. Letting the set of unknown rows at iteration $t$ to be $\bar{I}_{t-1}$ (with $\bar{I}_0 = [N]$). We draw a fresh sequence of vectors $\Psi^{(t)}$ from some distribution, we then recover some rows $\bar{I}_t^\complement \subseteq \bar{I}_{t-1}$ of $\Theta^*$ and store them in $\hat{\Theta}$.

1. Draw $\Psi^{(t)} = (\psi_{ik}^{(t)})_{k \in [K_t], i \in \bar{I}_{t-1}}$, we obtain $\theta_{ik}^* = e_i^\intercal \Theta^* \psi_{ik}^{(t)}$.
2. Consider the loss function

$$L(\Theta - \Theta^*, \Psi^{(t)}) := \frac{1}{K_t |\bar{I}_{t-1}|} \sum_{i \in \bar{I}_{t-1}} \sum_{k=1}^{K} |\langle \Theta_i, \psi_{ik}^{(t)} \rangle - \theta_{ik}^*|^2 .$$

3. Find a matrix $\Theta$ with rank $\leq r$ such that $L(\Theta - \Theta^*, \Psi^{(t)}) = 0$.
4. Initialize $\bar{I}_t \leftarrow \emptyset$.
5. For every $i \in \bar{I}_{t-1}$, draw $K$ fresh samples using $\tilde{\psi}_{i1}^{(t)}, \cdots, \tilde{\psi}_{iK}^{(t)}$ and compute $\sum_{k=1}^{K} |\langle \Theta_i, \tilde{\psi}_{ik}^{(t)} \rangle - \theta_{ik}^*|^2$. If $\sum_{k=1}^{K} |\langle \Theta_i, \tilde{\psi}_{ik}^{(t)} \rangle - \theta_{ik}^*|^2 > 0$ then add $i$ to $\bar{I}_t$ i.e., $\bar{I}_t \leftarrow \bar{I}_t \cup \{i\}$.
6. End routine when $\bar{I}_t = \emptyset$.

Suppose $\psi_{ik}$ are i.i.d random vectors such that there exist $\zeta, \xi > 0$ such that for any $x \in \mathbb{R}^d$, $\|x\| = 1$ we have:

$$\|\psi_{ik}\| \leq 1 \text{ almost surely}; \quad \mathbb{E}|\langle \psi_{ik}, x \rangle| \geq \frac{\zeta}{\sqrt{d}}; \quad \mathbb{E}\psi_{ik}\psi_{ik}^\intercal \leq \frac{1}{d\xi^2} \tag{2}$$

To give some intuition, the second condition above means that given any vector $x$, there is some overlap between the random vector $\psi$ and $x$, ensuring that every measurement gives us some information helping us to complete the matrix. The third assumption is a standard bound on the covariance matrix. Then we have the following theorem whose proof is presented in Section F.

**Theorem 4.** *Assume that $\sup_i \|\Theta_i^*\| \leq C_\theta$ and that the distribution of $\psi_{ik}^{(t)}$ satisfies equation 2. Suppose $K_t |\bar{I}_{t-1}| = C \frac{r|\bar{I}_{t-1}| + dr}{\zeta^2 \xi^2} \log \frac{d}{\zeta\xi} + C \frac{\log\left(\frac{\log N}{\delta}\right)}{\zeta^2 \xi^2}$. With probability at-least $1 - \delta$, the algorithm terminates after $\log N$ iterations and the output $\hat{\Theta}$ satisfies $\hat{\Theta} = \Theta^*$. Therefore, with probability at-least $1 - \delta$, the sample complexity for estimation of $\Theta^*$ is:*

$$C \frac{r(N + d \log N)}{\zeta^2 \xi^2} \log \frac{d}{\zeta\xi} + C \frac{\log N \log\left(\frac{\log N}{\delta}\right)}{\zeta^2 \xi^2}$$

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

# A  MORE DISCUSSION REGARDING POLICY SPACE

## A.1  NECESSITY OF RANDOMIZED POLICIES

We will first show that randomized policies might be necessary in such contexts with a simple example and show that obtaining states which satisfy conditions like equation 2 goes beyond simple reward maximization. Suppose $H = 1$, $\mathcal{S} = \{1\}$ and $\mathcal{A} = \{1, \ldots, d\}$. We consider the embedding $\psi(s, a) = e_a$. Suppose we want to obtain a policy $\pi$ such that whenever $S_1, A_1 \sim \pi$, $\lambda_{\min}(\mathbb{E}\psi(S_1, A_1)\psi(S_1, A_1)^\intercal)$ is maximized (where $\lambda_{\min}$ denotes the minimum eigenvalue). This is maximized when $\pi(da|s)$ is chosen to be the uniform distribution over $\mathcal{A}$ and the corresponding value is $1/d$. Note that whenever $\pi$ is a deterministic policy we will have $\lambda_{\min} = 0$ whenever $d > 1$. This is in contrast to reward maximization problems where, under general conditions, a deterministic optimal policy exists (See Theorem 1.7 in Agarwal et al. (2019)).

If fact, we can also show that the policy which minimizes $\|\mathbb{E}\psi(S_1, A_1) - \frac{1}{d}\sum_{a=1}^d e_a\|$ must also necessarily be random.

In the case of linear MDPs, we can find such a deterministic optimal policy $\Pi = (\pi_1, \ldots, \pi_H)$ as $\pi_h(s) = \arg\sup_a \langle \psi(s, a), u_h^* \rangle + \langle \phi(s, a), v_h^* \rangle$ (Jin et al., 2020b). This reduces the problem to estimating the parameters $u_h^*, v_h^*$ even when the state-action space is an infinite set. However, when such policies are not guaranteed to exist, as in case of functional maximization required in Section 6, the set of all policies can be intractably large. This is the justification for picking a nice enough policy space denoted by $\mathbb{Q}$.

## A.2  CONSTRUCTING POLICY SPACES

We consider any linear MDP satisfying the definition given in Section 2 and suppose $\mathcal{A}$ is finite. We consider the set of all probability distributions $\pi_h(a|s; u, v) \propto \exp(\langle \phi(s, a), u \rangle + \langle \psi(s, a), v \rangle)$. We consider $\mathbb{Q}_h = \{\pi_h(a|s, u, v) : u, v \in \mathcal{B}_d(R)\}$, We let our policy space be $\mathbb{Q} = \{\Pi = (\pi_1, \ldots, \pi_H) : \pi_h \in \mathbb{Q}_h\}$.

**Lemma 2.** *Consider the probability distribution over a finite set $[|\mathcal{A}|]$ give by $p_\beta(a) \propto \exp(\beta x_a)$ for every $a \in [|\mathcal{A}|]$ some $x_a \in \mathbb{R}^+$ and $\beta \in \mathbb{R}^+$. For any $\epsilon > 0$ and random variable $A \sim p_\beta$, we must have:*

$$\mathbb{P}(x_A < \sup_a x_a - \epsilon) \le |\mathcal{A}| \exp(-\beta\epsilon)$$

*And*

$$\mathbb{E}x_A \ge (\sup_a x_a - \epsilon)(1 - |\mathcal{A}| \exp(-\beta\epsilon))$$

**Lemma 3.** *Let $Q_h^*(s, a)$ be the optimal action-value function for the MDP. Then the policy $\Pi = (\pi_1, \ldots, \pi_h)$ given by $\pi_h(a|s) \propto \exp(\beta Q_h^*(s, a))$ is $\epsilon H + H^2|\mathcal{A}| \exp(-\beta\epsilon)$ sub-optimal for any $\epsilon > 0$*

*Proof.* Consider the optimal value function defined by $V_h^*(s) = \sup_a Q_h^*(s,a)$. Let $\bar{Q}_h(s,a)$ denote the optimal action value function under the policy $\Pi$ and let $\bar{V}(s) = \int \bar{Q}_h(s,a)\pi_h(da|s)$ denote the value at state $s$ with the policy $\Pi$. Clearly, we have: $\bar{Q}_H(s,a) = Q_H^*(s,a) = R(s,a)$. $\bar{Q}_h(s,a) \geq Q_h^*(s,a) - \eta$ uniformly. Then we have

$$
\begin{aligned}
\bar{V}_h(s) &\geq \int Q_h^*(s,a)\pi_h(da|s) - \eta \\
&\geq (\sup_a Q_h^*(s,a) - \epsilon)(1 - |\mathcal{A}|\exp(-\beta\epsilon)) - \eta \\
&\geq \sup_a Q_h^*(s,a) - \epsilon - H|\mathcal{A}|\exp(-\beta\epsilon) - \eta = V_h^*(s) - \epsilon - H|\mathcal{A}|\exp(-\beta\epsilon) - \eta \quad (3)
\end{aligned}
$$

In the second step, we have invoked Lemma 2. In the last step, we have used the fact that $Q_h^* \in [0, H]$ uniformly. Now, by the Bellman iteration, we have:

$$
\begin{aligned}
Q_{h-1}^*(s,a) &= R_{h-1}(s,a) + \mathbb{E}_{s' \sim P_{h-1}(|s,a)} V_h^*(s') \\
&\geq R_{h-1}(s,a) + \mathbb{E}_{s' \sim P_{h-1}(|s,a)} \bar{V}_h(s') - \epsilon - H|\mathcal{A}|\exp(-\beta\epsilon) - \eta \\
&= \bar{Q}_{h-1}(s,a) - \epsilon - H|\mathcal{A}|\exp(-\beta\epsilon) - \eta \quad (4)
\end{aligned}
$$

Therefore, by induction, we conclude that $\bar{V}_1(s) \geq V_1^*(s) - \epsilon H - H^2|\mathcal{A}|\exp(-\beta\epsilon)$

Therefore, by the definition of the value function, we conclude the claim. $\qquad\square$

Now, by a simple extension of Proposition 2.3 in Jin et al. (2020b), we conclude that the optimal $Q_h^*$ function for any linear MDP can be written as:

$$
Q_h^*(s,a) = \langle \psi(s,a), u_h^* \rangle + \langle \phi(s,a), v_h^* \rangle.
$$

Where $\|u_h^*\|_2 \leq \sqrt{d}$ and $\|v_h^*\|_\infty \leq HC_\mu$. Observe that choosing $\epsilon = \frac{\eta}{2H}$ and $\beta = 2\frac{\log(2H|\mathcal{A}|/\eta)}{\eta}$ will ensure that the randomized policy $\Pi$ in the statement of Lemma 3 is $\eta$ optimal. Therefore, we can take $R = 2dHC_\mu \frac{\log(2H|\mathcal{A}|/\eta)}{\eta}$ in the definition of $\mathbb{Q}_h$ above and conclude that this includes every $\eta$ optimal policy for every MDP with embedding functions $\phi, \psi$. We will now bound the covering number. Recall the definition of the distance $D_\mathbb{Q}(\Pi_1, \Pi_2) = \sup_{h \in [H]} \mathsf{TV}(\pi_h^{(1)}, \pi_h^{(2)})$. Therefore it is sufficient to obtain an $\eta$ cover for $\mathbb{Q}_h$ (denoted by $\hat{\mathbb{Q}}_{h,\eta}$) and then construct $\hat{\mathbb{Q}}_\eta = \{\Pi = (\pi_1, \ldots, \pi_H) : \pi_h \in \hat{\mathbb{Q}}_{h,\eta} \forall h \in [H]\} = \prod_{h=1}^H \hat{\mathbb{Q}}_{h,\eta}$.

**Lemma 4.** $\pi(|s; u, v)$ *be as defined in the beginning of this Subsection.*

$$
\mathsf{TV}(\pi(\cdot|s; u, v), \pi(\cdot|s; u', v')) \leq \frac{1}{2}\left(\exp(2\|u - u'\|_2 + 2\|v - v'\|_\infty) - 1\right)
$$

*Proof.* Denote $\pi(a|s; u, v)$ by $\pi(a)$ and $\pi(a|s; u', v')$ by $\pi'(a)$. Consider the corresponding partition functions denoted by $Z := \sum_{a \in \mathcal{A}} \exp(\langle \psi(s,a), u \rangle + \langle \phi(s,a), v \rangle)$ and $Z' := \sum_{a \in \mathcal{A}} \exp(\langle \psi(s,a), u' \rangle + \langle \phi(s,a), v' \rangle)$. We conclude that using Hölder's inequality for $\langle u - u', \psi \rangle$ and $\langle v - v', \phi \rangle$ that:

$$
\exp(-\|u - u'\|_2 - \|v - v'\|_\infty) \leq \frac{Z'}{Z} \leq \exp(\|u - u'\|_2 + \|v - v'\|_\infty)
$$

$$
\exp(-2\|u - u'\|_2 - 2\|v - v'\|_\infty) \leq \frac{\pi'(a)}{\pi(a)} \leq \exp(2\|u - u'\|_2 + 2\|v - v'\|_\infty) \quad (5)
$$

$$
\begin{aligned}
\mathsf{TV}(\pi, \pi') &= \tfrac{1}{2}\sum_{a \in \mathcal{A}} |\pi(a) - \pi'(a)| \\
&= \tfrac{1}{2}\sum_{a \in \mathcal{A}} \pi(a)\left|1 - \tfrac{\pi'(a)}{\pi(a)}\right| \\
&\leq \frac{1}{2}\left(\exp(2\|u - u'\|_2 + 2\|v - v'\|_\infty) - 1\right) \quad (6)
\end{aligned}
$$

$\qquad\square$

Using the lemma above, we conclude that $\hat{\mathbb{Q}}_{h,\eta} = \{\pi_h(\cdot|s; u, v) : u, v \in \hat{\mathcal{B}}_{d,\eta/4}(R)\}$ whenever $\eta \leq 1$. Here $\hat{\mathcal{B}}_{d,\eta/4}(R)$ an $\eta/4$ net over $\mathcal{B}_d(R)$ with respect to the norm $\|\cdot\|_2$. From the results in Vershynin (2018), we can therefore take:

$$|\hat{\mathbb{Q}}_{h,\eta}| \leq |\hat{\mathcal{B}}_{d,\eta/4}(R)|^2 \leq \exp(Cd\log(\tfrac{C\eta}{R})) \tag{7}$$

Since we had $\hat{\mathbb{Q}}_\eta = \prod_{h=1}^H$, we conclude that:

$$\log(|\hat{\mathbb{Q}}_\eta|) \leq cdH\left(\log\tfrac{dH}{\eta} + \log\log(2H|\mathcal{A}|/\eta)\right)$$

### A.3 RELATIONSHIP TO MEANFIELD LIMITS OF MULTI-AGENT RL

Consider the conditions given in equation 2 for $\psi(S_h, A_h)$ where $S_{1:H}, A_{1:H} \sim \Pi^{\mathsf{mat-comp},h}$. We refer to the proof of Theorem 3 to show that the following condition implies the conditions given in equation 2:

$$\inf_{x\in\mathcal{S}^{d-1}} \mathbb{E}|\langle\psi(S_h, A_h), x\rangle|\sqrt{d} - \xi d\langle\psi(S_h, A_h)\rangle^2 \geq \zeta$$

Conversely, the conditions in equation 2 implies the following:

$$\inf_x \mathbb{E}|\langle\psi(S_h, A_h), x\rangle|\sqrt{d} - \frac{\xi^2\zeta}{2}d\langle\psi(S_h, A_h), x\rangle^2 \geq \frac{\zeta}{2}$$

Therefore, the problem of obtaining a policy satisfying equation 2 reduces to finding a policy such that the functional $\inf_x \mathbb{E}|\langle\psi(S_h, A_h), x\rangle|\sqrt{d} - \beta d\langle\psi(S_h, A_h), x\rangle^2$ is maximized for some small enough $\beta \in \mathbb{R}^+$. Note that this functional maps the distribution of $(S_h, A_h)$ (denoted by $\Gamma_h(\Pi)$) obtained by applying some policy $\Pi$ to a real number. Let us denote this function by $J(\Gamma_h)$. Our objective now is to find a policy $\Pi^{\mathsf{mat-comp},h}$ by solving the following optimization problem

$$\arg\sup_{\Pi\in\mathbb{Q}} J(\Gamma_h(\Pi)) \tag{8}$$

This setup is similar to the mean field multi-agent control problem presented in Cammardella et al. (2020). To explicitly see the connection to Multi-agent systems, consider $n$ agents with the same MDP $\mathcal{M}$ and embedding functions $\phi, \psi$. Each trajectory from this multi-agent system corresponds to jointly and running the MDP associated with each agent with the same policy independently. The collective reward of the system is given by $J(\hat{\Gamma}_h)$, where $\hat{\Gamma}_h$ denotes the empirical distribution of state-actions of the $n$ agents at time $h$. Note that on the one hand, picking a policy $\Pi_n$ to maximize this reward is a reward maximization problem on the joint multi-agent system. And, for any fixed policy $\Pi$, as $n \to \infty$, $\hat{\Gamma}_h(\Pi) \to \Gamma_h(\Pi)$ under reasonable assumptions on the state space via the law of large numbers and hence $J(\hat{\Gamma}_h(\Pi)) \to J(\Gamma_h(\Pi))$ under continuity. Therefore the planning problem in equation 8 is the same as the multi-agent planning problem described above in the limit $n \to \infty$.

## B ANALYSIS - TABULAR MDPS

We will call the reward free RL procedure in Phase 1 to be successful if it outputs the $\epsilon$ optimal policy. This has probability atleast $1 - \frac{\delta}{2}$.

### B.1 ANALYSIS OF ALGORITHM 1

**Lemma 5.** *Suppose $p \leq \frac{1}{2}$, conditioned on the success of Phase 1, with probability at-least $1 - \exp(-cNp|\mathcal{S}||\mathcal{A}|H)$, Algorithm 1 terminates after querying $\frac{C|\mathcal{S}||\mathcal{A}|NHp}{\epsilon}$ trajectories. $(\mathcal{G}_h)_{h\in[H]}$, the active sets at the termination of the algorithm. They satisfy:*

$$\sup_\pi \sum_{h=1}^H P_h^\pi(\mathcal{G}_h) \leq \frac{5\epsilon}{8} \tag{9}$$

For any $a \times b$ matrix $R$, let $\hat{R}$ be its partially observed version (that is, there exists a set of indices $I \subseteq [a] \times [b]$ such that $\hat{R}_{ij} = R_{ij}$ if $(i,j) \in I$ and $\hat{R}_{ij} = *$ otherwise). We call a random set of indices $J$ to have the distribution $\mathsf{Unif}(m, [a], [b])$ if $J$ is drawn uniformly at random such that $|J| = m$.

**Lemma 6** (Modification: Mod1). *Suppose we run, independently, a modification of algorithm 1 where on the "Query trajectory" step the trajectories are sampled from a fixed MDP $\mathcal{M}_1$ (but rewards are from the reward function corresponding to $U_t$). Consider all the random variables that determine the trajectory of this algorithm: $\left(\hat{V}, \hat{\Pi}, (S_{1:H}^{(t)}, A_{1:H}^{(t)}, R_{1:H}^{(t)})_t, (U_t)_t\right)$. Then the joint distribution of this collection of random variables is unchanged under the modification.*

*Proof.* The proof follows from an induction argument on the time index $t$. We describe the key steps here. For the ease of notation, let $\mathcal{T}_t = \left((S_{1:H}^{(t)}, A_{1:H}^{(t)}, R_{1:H}^{(t)}), U_t\right)$ Let $X_T = \left(\hat{V}, \hat{\Pi}, (\mathcal{T}_t)_{t \leq T}\right)$. Let $\tilde{X}_T$ and $\tilde{\mathcal{T}}_t$ denote the corresponding quantities under the modification. It is enough to show that finite dimensional marginals have the same joint distribution under the modification. In particular, we will show:

1. $\mathcal{T}_0 \overset{d}{=} \tilde{\mathcal{T}}_0$

2. Suppose $X_T \overset{d}{=} \tilde{X}_T$. Then the Markov kernel $k_{\mathcal{T}_{T+1}|X_T}$ is almost surely (under the common distribution of $X_T, \tilde{X}_T$) equal to $k_{\tilde{\mathcal{T}}_{T+1}|\tilde{X}_T}$. Thus $X_{T+1} \overset{d}{=} \tilde{X}_{T+1}$

The first statement is straightforward since, in the zeroth step, the distribution of $\hat{\Pi}^{\mathcal{G}}$ not affected by the modification, and thus due to identical MDP transitions across users, the distribution of $\mathcal{T}_0$ is preserved under modification. A similar argument proves the second statement. Roughly, given a realization of $X_T$ the distribution of $\mathcal{T}_{T+1}$ is same as the distribution of $\tilde{\mathcal{T}}_{T+1}$ given the same realization of $\tilde{X}_T$, due to the exact same reason presented for the first statement. A fully formal proof requires setting up appropriate proability spaces, so we omit it here. Furthermore, since the random variables considered are all discrete, one can argue via PMFs as well. $\square$

**Lemma 7.** *Suppose $p \leq \frac{1}{2}$. conditioned on the success of Phase 1 and termination of Algorithm 1, for every $h \in [H]$, the Algorithm 1 returns partially filled reward matrices $\hat{R}_h$. Consider the sub-matrix $\hat{R}_h^{\mathcal{G}_h^{\complement}}$. Let $I_h \subseteq [N] \times \mathcal{G}_h^{\complement}$ be the sub-set of observed indices for $\hat{R}_h$. Let $J_h|\mathcal{G}_h \sim \mathsf{Unif}(\frac{Np|\mathcal{G}_h^{\complement}|}{2}, [N], \mathcal{G}_h^{\complement})$. There exists a coupling between $J_h$ and $I_h$ such that*

$$\mathbb{P}\left(J_h \subseteq I_h | \mathcal{G}_h\right) \geq 1 - |\mathcal{S}||\mathcal{A}| \exp(-cNp)$$

*Proof.* Let us fix $\mathcal{G}_h$ and construct a coupling between $I_h$ and $J_h$.

Consider any fixed, arbitrary permutations $\sigma_g$ over $[N]$, for $g \in \mathcal{G}_h^{\complement}$. By $\sigma(I_h)$, we denote $\{(\sigma_g(i), g) : (i, g) \in I_h\}$.

**Claim 1.** *Conditioned on $\mathcal{G}_h$, $\sigma(I_h)$ has the same distribution as $I_h$.*

*Proof.* Let $\{\sigma_{(s,a):[N]\rightarrow[N]}|(s,a) \in \mathcal{S} \times \mathcal{A}\}$ be a set of arbitrary permutations on $[N]$. From lemma 6 it is enough to prove the statement for the random variables under the modification described in that lemma (call this Mod1). Now consider a further modification (call it Mod2) where in every iteration $t$, we sample $U_t \sim \mathsf{Unif}([N])$, for each horizon $h$ we set $\tilde{U}_h^{(t)} = \sigma_{(S_h^{(t)}, A_h^{(t)})}(U_t)$, and then update the entries of $R_h(\tilde{U}_h^{(t)}, (S_h^{(t)}, A_h^{(t)}))$ (instead of $R_h(U_t, (S_h^{(t)}, A_h^{(t)}))$). Next, we couple these two modifications by using same $(\hat{V}, \hat{\Pi})$ and the same set of $U_t$'s for both the modifications. Further, we couple the MDP used in these modifications to be the same, single MDP.

Now an induction argument shows that the sequence of active sets $\mathcal{G}$ obtained in these modifications are also identical for every time $t$; only the rows of $R_h$ where entries are filled change according to the set of permutations chosen.Thus, under the described coupling, Mod1 and Mod2 produce

identical trajectories (i.e., $(S_{1:H}^{(t)}, A_{1:H}^{(t)})$), the columns of reward matrices are just permutations of each other described by the chosen set of permutations, and algorithm 1 terminate at the same time in both these cases. However, the same induction argument also shows that for each $t$ and $h$, conditioned on $\mathcal{G}_h$, trajectories (which is same in Mod1 and Mod2) until at beginning of iteration $t$, we have $(U_t, (S_h^{(t)}, A_h^{(t)})) \stackrel{d}{=} (\tilde{U}_h^{(t)}, (S_h^{(t)}, A_h^{(t)}))$.

Therefore if $I_h, \tilde{I}_h \subset [N] \times \mathcal{G}_h^\complement$ denotes the subset of observed indices at termination (outside the active set), then $\tilde{I}_h = \sigma(I_h) \equiv \{(\sigma_{(s,a)}(i), (s,a)) : (i, (s,a)) \in I_h)\}$ and, conditioned on $\mathcal{G}_h$, $\tilde{I}_h \stackrel{d}{=} I_h$

$\square$

**Claim 2.** *At termination, conditioned on $\mathcal{G}_h$, random sets $I_h^g = \{(i, g) : (i, g) \in I_h\}$ are jointly independent.*

*Proof.* Again we work with the modification Mod1 described in lemma 6. For each $(s, a)$ consider the collection of $U_t$'s that are used populate the column $(s, a)$ of matrix $R_h$ in algorithm 1. Call this collection $\mathcal{U}_{(s,a)}$. $\square$

**Remark 3.** *Since the columns of $I_h$ have exactly $Np$ entries, permutation invariance proved in the above claim implies that*

For any set $\bar{J} \subseteq [N] \times \mathcal{G}_h^\complement$, define the count function $(N_g^{\bar{J}})_{g \in \mathcal{G}_h^\complement}$ such that $N_g^{\bar{J}} = |\{i \in [N] : (i, g) \in \bar{J}\}|$.

We are now ready to give the coupling: given $\mathcal{G}_h$, draw uniformly random, independent permutations $\sigma_g$ for $g \in \mathcal{G}_h^\complement$. Draw $(N_g)_{g \in \mathcal{G}_h^\complement}$ independent of $\sigma_g$ and to have the joint law of $(N_g^{J_h})_{g \in \mathcal{G}_h^\complement}$. Define:

$$\tilde{J}_h = \{(\sigma_g(i), g) : i \le N_g, g \in \mathcal{G}_h^\complement\}$$

$$\tilde{I}_h = \{(\sigma_g(i), g) : i \le Np, g \in \mathcal{G}_h^\complement\}$$

**Claim 3.** *The marginal distributions of $\tilde{J}_h$ and $\tilde{I}_h$ are respectively the distributions of $J_h$ and $I_h$.*

*Proof.* First we will prove a general statement about $J \sim \mathsf{Unif}(r, [N], [M])$. Let $X \in \{0, 1\}^{N \times M}$ with $X_{i,m} = 1$ iff $(i, m) \in J$. Let $(N_m)_{m \in [M]}$ be the count functions corresponding to $J$ i.e., $N_m = \sum_i X_{i,m}$. Let $Y_m = (X_{1,m}, \cdots, X_{N,m})$. We will argue that conditional on $\{N_m : m \in [M]\}$, the random vectors $Y_m$ are jointly independent. Indeed, pick any $x \in \{0, 1\}^{N \times M}$ and $(n_1, \cdots, n_m)$. Let $y_m$ be the $m$'th column of $x$. Then

$$\mathbb{P}[X = x, \cap_m \{N_m = n_m\}] = \left(\prod_m \mathbb{1}\left[\sum_i x_{i,m} = n_m\right]\right) \mathbb{1}\left[\sum_{i,m} x_{i,m} = r\right] \frac{1}{\binom{MN}{r}}$$

The above can also be written as

$$\mathbb{P}[\cap_m \{Y_m = y_m\}, \cap_m \{N_m = n_m\}] = \left(\prod_m \mathbb{1}\left[\sum_i x_{i,m} = n_m\right]\right) \mathbb{1}\left[\sum_m n_m = r\right] \frac{1}{\binom{MN}{r}}$$

Let $\mathbf{1}$ denote the all 1 vector in $\mathbb{R}^N$. Note that $y_m = (x_{1,m}, \cdots, x_{N,m})^\top$. Marginalizing the above, we see that

$$\mathbb{P}[\cap_m \{N_m = n_m\}] = \left(\prod_m \frac{1}{\binom{N}{n_m}}\right) \mathbb{1}\left[\sum_m n_m = r\right] \frac{1}{\binom{MN}{r}}$$

Thus the conditional distribution can be expressed as

$$\mathbb{P}\left[\cap_m \{Y_m = y_m\} \,\middle|\, \cap_m \{N_m = n_m\}\right] = \left(\prod_m \frac{\mathbb{1}\left[\mathbf{1}^\top y_m = n_m\right]}{\binom{N}{n_m}}\right) \frac{\mathbb{1}\left[\sum_m n_m = r\right]}{\binom{MN}{r}}$$

Since the (conditional) joint PMF factors, it is an easy calculation to show the conditional independence i.e.,

$$\mathbb{P}\left[\cap_m \{Y_m = y_m\} \middle| \cap_m \{N_m = n_m\}\right] = \prod_m \mathbb{P}\left[Y_m = y_m \middle| \cap_m \{N_m = n_m\}\right]$$

Furthermore, for any $n_1, \cdots n_m$ such that $\sum_m n_m = r$, marginalization shows

$$\mathbb{P}\left[Y_m = y_m | \cap_m \{N_m = n_m\}\right] = \frac{1\left[\mathbf{1}^\top y_m = n_m\right]}{\binom{N}{n_m}}$$

Let $N_{-m} = (N_1, \cdots, N_{m-1}, N_{m+1}, \cdots, N_M)$, and similarly for $n_{-m}$. Then

$$\mathbb{P}\left[Y_m = y_m, N_{-m} = n_{-m} | N_m = n_m\right]$$
$$= \mathbb{P}\left[Y_m = y_m | \cap_m \{N_m = n_m\}\right] \mathbb{P}\left[N_{-m} = n_{-m} | N_m = n_m\right]$$
$$= \begin{cases} 0, & \sum_m n_m \neq r \\ \frac{1\left[\mathbf{1}^\top y_m = n_m\right]}{\binom{N}{n_m}} \mathbb{P}\left[N_{-m} = n_{-m} | N_m = n_m\right], & \text{otherwise} \end{cases}$$

The above factorization directly implies that $Y_m$, conditioned on $N_m$ is uniformly distributed on its support $\{y : \mathbf{1}^\top y = N_m\}$ and is independent of $N_{-m}$. Thus

$$\mathbb{P}\left[\cap_m \{Y_m = y_m\} \middle| \cap_m \{N_m = n_m\}\right]$$
$$= \prod_m \mathbb{P}\left[Y_m = y_m \middle| N_m = n_m\right]$$
$$= \prod_m \frac{1\left[\mathbf{1}^\top y_m = n_m\right]}{\binom{N}{n_m}}$$

**Observation**: The above calculations give another way to generate $Y$: first generate $N_1, \cdots, N_M$ from the right distribution, and then conditioned on $N_m$ generate each $Y_m$ uniformly such that $\mathbf{1}^\top Y_m = N_m$.

Next we apply the above calculations and observation to $J = J_h | \mathcal{G}_h \sim \mathsf{Unif}(\frac{Np|\mathcal{G}_h^{\complement}|}{2}, [N], \mathcal{G}_h^{\complement})$. For a uniformly random permutation $\sigma$ on $[N]$, the set $\{\sigma(i) : 1 \leq i \leq k\}$ is uniformly distributed on all $k$-sized subsets of $[N]$. In the statement of the claim the permutations are chosen independently for each $g \in \mathcal{G}_c^{\complement}$. Thus from the above observation, we have $J_h \overset{d}{=} \tilde{J}_h$ conditioned on $\mathcal{G}_h$.

The claim about $\tilde{I}_h$ follows directly from permutation invariance proved by claim 1.

$\square$

**Claim 4.** $\mathbb{P}(N_g > Np | \mathcal{G}_h) \leq \exp(-c_0 Np)$ *for every* $g \in \mathcal{G}_h^{\complement}$

*Proof.* Throughout this proof, we will condition on the terminal active set $\mathcal{G}_h$. We will show this using the results on concentration with negative regression property as established in Proposition 29 in Dubhashi & Ranjan (1996). $N_g = \sum_{i=1}^N \mathbb{1}((i,g) \in \tilde{J}_h)$. Now we will show that the collection $X_{ig} := \mathbb{1}((i,g) \in \tilde{J}_h)$ for $i \in [N], g \in \mathcal{G}_h^{\complement}$ satisfy the negative regression property. By the definition of negative regression, we can conclude that the sub-collection $(X_{ig})_{i \in [N]}$ also satisfies this property for every $g \in \mathcal{G}_h^{\complement}$.

Consider the partial order over binary vectors $X \succeq Y$ iff $X_l \geq Y_l$ for every $l$. The negative regression property is satisfied iff for every $K_1, K_2 \subseteq [N] \times \mathcal{G}_h^{\complement}$ such that $K_1 \cap K_2 = \emptyset$, and a real valued function $f(X_m : m \in K_1)$ which is non-decreasing with respect to the partial order, we must have:

$$g(t_l : l \in K_2) := \mathbb{E}\left[f(X_m : m \in K_1) \middle| X_l = t_l, \forall l \in K_2\right]$$

be such that $g$ is a non-increasing function in $t_l$ with respect to the partial order. Note that in the case of uniform distribution as in $\tilde{J}_h$, the distribution $(X_m)_{m \in K_1}$ is the uniform, permutation invariant distribution with constant sum almost surely. The sum being $\frac{Np|\mathcal{G}_h^\complement|}{2} - \sum_{l \in K_2} t_l$. Therefore, whenever $t_l' \geq t_l$ for every $l \in K_2$, we have the following stochastic dominance:

$$\left[ (X_m)_{m \in K_1} \Big| X_l = t_l' \forall l \in K_2 \right] \preceq \left[ (X_m)_{m \in K_1} \Big| X_l = t_l \forall l \in K_2 \right]$$

Therefore, this coupling leads us to conclude that:

$$\begin{aligned} g(t_l : l \in K_2) &:= \mathbb{E}\left[ f(X_m : m \in K_1) \big| X_l = t_l \forall l \in K_2 \right] \\ &\geq \mathbb{E}\left[ f(X_m : m \in K_1) \big| X_l = t_l' \forall l \in K_2 \right] \\ &= g(t_l' : l \in K_2) \end{aligned} \tag{10}$$

The second step follows from stochastic dominance. This implies that the function $g$ is non-increasing which establishes the negative regression property. Now, we consult Proposition 29 in Dubhashi & Ranjan (1996) to show that we can take Chernoff bounds on $N_g = \sum_{i \in [N]} X_{ig}$ as though $X_{ig}$ were i.i.d $\mathsf{Ber}(p)$. Therefore, from an application of Bernstein's inequality (Boucheron et al., 2013), we conclude the statement of the claim. □

Now, $J_h \subseteq I_h$ if and only if $N_g \leq Np$ for every $g \in \mathcal{G}_h^\complement$. Therefore, from the claim above, we have $\mathbb{P}(J_h \subseteq I_h | \mathcal{G}_h^\complement) \geq 1 - |\mathcal{S}||\mathcal{A}| \exp\left(-c_0 Np\right)$. □

We are now ready to prove Theorem 1.

*Proof of Theorem 1.* In order to establish the result, we need to show that with $p$ as set in the statement, the algorithm returns $\epsilon$ optimal policies $\hat{\Pi}_u$ for every user $u \in [N]$ with probability at-least $1 - \delta$.

The total sample complexity is the number of trajectories queried in Phase 1 plus the number of trajectories queried in Phase 2. Phase 1 queries $K^{\mathsf{rf}}(\frac{\epsilon}{8}, \frac{\delta}{2})$ trajectories, which is $C \frac{|\mathcal{S}||\mathcal{A}|H^2\left(|\mathcal{S}| + \log(\frac{1}{\delta})\right)}{\epsilon^2} \mathsf{polylog}(\frac{|\mathcal{S}||\mathcal{A}|H}{\epsilon})$ by the results of Zhang et al. (2020). By Lemma 5, we conclude that the sample complexity of phase 2 is $\frac{C|\mathcal{S}||\mathcal{A}|NHp}{\epsilon}$ and with the value of $p$ given in the statement of the theorem, this succeeds with probability at-least $1 - \frac{\delta}{4}$ when conditioned on the success of Phase 1.

We will show that conditioned on the success of Phase 2, with probability at-least $1 - \frac{\delta}{4}$, the nuclear norm minimization algorithm of Recht (2011) successfully obtains $R_h^{\mathcal{G}_h^\complement}$. Indeed by Theorem 1 in Recht (2011), we see that whenever co-ordinates of $R_h^{\mathcal{G}_h^\complement}$ corresponding to random indices drawn from $\mathsf{Unif}(m, [N], \mathcal{G}_h^\complement)$ are observed with $m = C_1 \max(\mu_1^2, \mu_0) r(N + |\mathcal{G}_h^\complement|) \log^2 |\mathcal{G}_h^\complement| \log(\frac{H}{\delta})$, the algorithm succeeds at recovering $R_h^{\mathcal{G}_h^\complement}$ with probability at-least $1 - \frac{\delta}{8H}$. The number of co-ordinates we observe is

$$Np|\mathcal{G}_h^\complement| \geq \frac{Np|\mathcal{S}\mathcal{A}|}{2} \geq 2C_1 \max(\mu_1^2, \mu_0) r(N + |\mathcal{G}_h^\complement|) \log^2 |\mathcal{G}_h^\complement| \log(\frac{H}{\delta})$$

In the last step, we have used Assumption 2 to conclude that $|\mathcal{G}_h^\complement| \geq \frac{|\mathcal{S}||\mathcal{A}|}{2}$. For the constant $C$ in the definition of $p$ large enough, we must have:

$$m \leq \frac{Np|\mathcal{G}_h^\complement|}{2}$$

Note that the results of Recht (2011) requires at-least $m$ observations to be chosen uniformly at random co-ordinates, but we do not obtain observations which are uniformly at uniformly random co-ordinates. Here, we will use the results of Lemma 7. Let $J_h$ be a fictitious subset of co-ordinates

distributed as $\mathsf{Unif}(m, [N], \mathcal{G}_h^{\complement})$ when conditioned on $\mathcal{G}_h^{\complement}$. If the observed co-ordinates are $J_h$, then we can successfully estimate the reward matrix $R_h$ with proability at-least $1 - \frac{\delta}{8H}$ in this case. Now, suppose that the actually observed co-ordinates are $I_h$, which is a strict super-set of $J_h$. Then we check that the matrix completion algorithm, which is based on constrained nuclear-norm minimization, still succeeds with observed co-ordinates corresponding to $I_h$ whenever it succeeds with the observed co-ordinates correspond to $J_h$.

We now refer to the coupling in Lemma 7, which shows that we can couple $J_h$ to the real distribution $I_h$ such that $J_h \subseteq I_h$ with probability at-least $1 - \frac{\delta}{8H}$ When the constant $C_1$ in the definition of $p$ is large enough, we conclude by invoking Lemma 7 that: $J_h \subseteq I_h$ with probability at-least $1 - \frac{\delta}{8H}$. Applying union bound for $h \in [H]$, we conclude that Phase 3 succeeds with probability at-least $1 - \frac{\delta}{4}$ when conditioned on the success of Phases 1 and 2.

Therefore, from the arguments above, we conclude that Phases 1,2 and 3 succeed with probability at-least $1 - \delta$ and give us the reward matrices $R_h^{\mathcal{G}_h^{\complement}}$ where the sets satisfy the following equation from Lemma 5.

$$\sup_\pi \sum_{h=1}^{H} P_h^\pi(\mathcal{G}_h) \leq \frac{5\epsilon}{8} \tag{11}$$

It now remains to show that we obtain $\epsilon$ optimal policies for each user after Phase 4. Note that whenever Phase 1 succeeds, we can compute $\epsilon/4$ optimal policies for every possible reward function bounded in $[0,1]$. Since we do not know the rewards over the set $\mathcal{G}_h$, we set it to zero as described in the algorithm to obtain $\bar{R}_h$. It remains to show that planning with $\bar{R}_h$ and using it with the reward free RL algorithm gives us an $\epsilon$ optimal policy. Suppose $\Pi_u^*$ is the optimal policy for user $u$ and suppose $\bar{\Pi}_u$ be the optimal policy for user $u$ under rewards $\bar{R}_h(u, (s,a))$. Note that combined with the guarantees for the reward free RL, in order to complete the proof of the theorem, it is sufficient to show that the policy $\bar{\Pi}_u$ is $3\epsilon/4$ optimal with respect to the actual rewards $R_h(u, (s,a))$. Let $S_{1:H}^*, A_{1:H}^* \sim \mathcal{M}(\Pi_u^*)$ and $\bar{S}_{1:H}, \bar{A}_{1:H} \sim \mathcal{M}(\bar{\Pi}_u)$.

$$\mathbb{E} \sum_{h=1}^{H} R_h(u, (S_h^*, A_h^*)) \leq \mathbb{E} \sum_{h=1}^{H} R_h(u, (S_h^*, A_h^*)) \mathbb{1}((S_h^*, A_h^*) \in \mathcal{G}_h^{\complement}) + \mathbb{1}((S_h^*, A_h^*) \in \mathcal{G}_h)$$

$$= \mathbb{E} \sum_{h=1}^{H} \bar{R}_h(u, (S_h^*, A_h^*)) + \mathbb{1}((S_h^*, A_h^*) \in \mathcal{G}_h) = \mathbb{E} \sum_{h=1}^{H} \bar{R}_h(u, (S_h^*, A_h^*)) + \sum_{h=1}^{H} P_h^{\Pi^*}(\mathcal{G}_h)$$

$$\leq \mathbb{E} \sum_{h=1}^{H} \bar{R}_h(u, (S_h^*, A_h^*)) + \frac{5\epsilon}{8}$$

$$\leq \mathbb{E} \sum_{h=1}^{H} \bar{R}_h(u, (\bar{S}_h, \bar{A}_h)) + \frac{5\epsilon}{8}$$

$$\leq \mathbb{E} \sum_{h=1}^{H} R_h(u, (\bar{S}_h, \bar{A}_h)) + \frac{5\epsilon}{8}$$

$$\tag{12}$$

In the first step we have used the fact that the rewards are uniformly bounded in $[0,1]$. In the second step, we have used the definition of $\bar{R}_h(u, (s,a)) := R_h(u, (s,a)) \mathbb{1}((s,a) \in \mathcal{G}_h^{\complement})$. In the third step, we have used the guarantee in equation 11. In the fourth step, we have used the fact that $\bar{P}i$ maximizes the reward $\bar{R}_h$. In the fifth step, we have used the fact that $R_h(u, (s,a)) \geq \bar{R}_h(u, (s,a))$ uniformly. From the discussion above, this concludes the proof of the theorem. $\square$

## C   ANALYSIS - LINEAR MDPS

**Lemma 8.** *Suppose Assumption 2 holds. Let $\kappa > 1$ and $T \geq C\frac{d\kappa^2}{(\gamma-\epsilon)^2}\log\frac{d\kappa}{\gamma-\epsilon}$. With probability at-least $1 - H\exp(-c(\gamma-\epsilon)T)$, the output of Algorithm 2 returns $\phi_{th}$ such that $\sum_{t=1}^{T}\phi_{th}\phi_{th}^{\mathsf{T}} \succeq \kappa^2 I$ for every $h \in [H]$*

*Proof of Theorem 2.* By Theorem 1 in Wagenmaker et al. (2022), we take $K^{\mathsf{rf}}(\epsilon, \delta/4) = \frac{CdH^5(d+\log(\frac{1}{\delta}))}{\epsilon^2} + \frac{Cd^{9/2}H^6\log^4(\frac{1}{\delta})}{\epsilon}$. Phase 1 succeeds with probability $1 - \frac{\delta}{4}$.

Note that this is the quantity $T_{\mathsf{rf}}$ in the statement of the theorem. We now condition on the success of Phase 1. The number of trajectories queried by Algorithm 2 which is given by $HT = T_{\mathsf{pol}}$. By Lemma 8, we conclude that for the given values of $T$ and $\kappa$, this algorithm successfully outputs $\phi_{ht}$ such that $G_{\phi,h} \succeq \kappa^2 I$ for every $h \in [H]$, with probability at-least $1 - \frac{\delta}{4}$.

Now, condition on the success of Algorithm 2. By theorem 3, we conclude that with these conclude that with proabability at-least $1 - \frac{\delta}{4}$, with the values of the given parameters, for every $h \in [H]$, the procedure in Step 2 of Phase 2 outputs a policy $\hat{\Pi}^{f,h}$ such that whenever $S_{1:H}, A_{1:H} \sim \mathcal{M}(\hat{\Pi}^{f,h})$, the conditions in equation 2 is satisfied for the random vector $\psi(S_h, A_h)$ with $\zeta$ replaced by $\zeta/2$. We then use the active learning based matrix completion procedure given in Section 6, where the vectors $\psi_{jk}$ are sample using the policy $\hat{\Pi}^{f,h}$ on the given user. By theorem 4, we conclude that conditioned on the success of all the steps above, with probability $1 - \frac{\delta}{4}$, we can exactly estimate each of the matrices $\Theta_h^*$ for $h \in [H]$ with $T_{\mathsf{mat-comp}}$ number of samples.

Upon the success of Phases 1, 2, 3 (which occurs with probability at-least $1 - \delta$ by union bound), we conclude that Phase 4 gives the $\epsilon$ optimal policy for each user $u \in [N]$ because of the guarantees of reward free RL.

$\square$

## D   DEFERRED PROOFS

### D.1   PROOF OF LEMMA 5

*Proof.* We suppose that the reward free RL in Phase 1 succeeds and returns the $\frac{\epsilon}{8}$ optimal policy for every choice of rewards bounded in $[0, 1]$. The algorithm terminates whenever the active sets are such that

$$\hat{V}(\mathcal{J}(;\mathcal{G})) \leq \frac{\epsilon}{2} \tag{13}$$

Note that by the definition of $\mathcal{J}(;\mathcal{G})$, the maximum value for the MDP with reward $\mathcal{J}(;\mathcal{G})$ is $\sup_\Pi \sum_{h=1}^{H} P^\Pi(\mathcal{G}_h)$. Since $\hat{V}$ is the output of the reward free RL algorithm, we conclude that we have:

$$|\hat{V}(\mathcal{J}(;\mathcal{G})) - \sup_\Pi \sum_{h=1}^{H} P_h^\Pi(\mathcal{G}_h)| \leq \frac{\epsilon}{8} \tag{14}$$

We conclude via equation 13 and equation 14 that equation 9 holds, which establishes the second part of the theorem. We now consider the termination time.

Suppose $\mathcal{G}^{(t)}$ is the sequence of active sets before termination at step $t$ (i.e, it satisfies $\hat{V}(\mathcal{J}(;\mathcal{G}^{(t)})) > \frac{\epsilon}{2}$). Recall $\hat{\Pi}$, the output of the reward free RL algorithm. It follows from the guarantees for reward free RL that:

$$|\sum_{h=1}^{H} P_h^{\hat{\Pi}^\mathcal{G}}(\mathcal{G}_h^{(t)}) - \sup_\Pi \sum_{h=1}^{H} P_h^\Pi(\mathcal{G}_h^{(t)})| \leq \frac{\epsilon}{8}$$

Combining this with Equation equation 14 and the fact that $\hat{V}(\mathcal{J}(;\mathcal{G}^{(t)})) > \frac{\epsilon}{2}$, we conclude:

$$\sum_{h=1}^{H} P_h^{\hat{\Pi}^\mathcal{G}}(\mathcal{G}_h^{(t)}) \geq \frac{\epsilon}{4} \tag{15}$$

We consider the potential function with $\varphi(0) = 0$ and $\varphi(t) = \sum_{h \in [H]} \sum_{(s,a) \in \mathcal{S} \times \mathcal{A}} T^{(t)}_{h,(s,a)}$, where $T^{(t)}_{h,(s,a)}$ is the counter $T_{h,(s,a)}$ inside Algorithm 1 at the beginning of the step $t$.

Whenever $\mathcal{G}^{(t)}$ is such that $\hat{V}(\mathcal{J}(; \mathcal{G}^{(t)})) > \frac{\epsilon}{2}$, we define $N_t := \varphi(t+1) - \varphi(t)$ (i.e, before termination). Just for the sake of theoretical arguments, we define the fictious random variables $N_t = \text{Ber}(\frac{\epsilon}{8})$ i.i.d after termination. Let $\mathcal{F}_t = \sigma(\mathcal{G}^{(s)}, S^{(s)}_{1:H}, A^{(s)}_{1:H}, R^{(s)}_{1:H}, U^{(s)} : s \leq t)$

**Claim 5.** *The following relations hold:*

1. $\mathbb{E}[N_t | \mathcal{F}_t] \geq \frac{\epsilon}{8}$

2. $\mathbb{E}[N_t^2 | \mathcal{F}_t] \leq \frac{\mathbb{E}[N_t | \mathcal{F}_t] H}{4}$

3. $|N_t| \leq H$ *almost surely.*

*Proof.* The inequalities are clear when $\mathcal{G}^{(t)}$ is such that $\hat{V}(\mathcal{J}(; \mathcal{G}^{(t)})) \leq \frac{\epsilon}{2}$. Now consider the case $\hat{V}(\mathcal{J}(; \mathcal{G}^{(t)})) > \frac{\epsilon}{2}$. By definition, conditioned on this event, we have almost surely:

$$N_t = \sum_{h=1}^H \mathbb{1}((S^{(t)}_h, A^{(t)}_h) \in \mathcal{G}^{(t)}_h) . \mathbb{1}(\hat{R}^{(t)}_h(U_t, (S^{(t)}_h, A^{(t)}_h)) = *)$$

That is, we increment the $T_{h,(s,a)}$ only when we encounter an element of the active set such that the entry for this user has not been observed before. Observe that for any arbitrary $(s,a) \in \mathcal{S} \times \mathcal{A}$

$$\mathbb{P}\left(\hat{R}^{(t)}_h(U_t, (s,a)) = * \Big| \mathcal{F}_t, (S^{(t)}_h, A^{(t)}_h) = (s,a)\right) = \frac{|\{u : \hat{R}^{(t)}_h(s,a) = *\}|}{N}. \tag{16}$$

This is true since the law of $S^{(t)}_h, A^{(t)}_h$ is independent of $U_t$ (since all users share the same MDP), when conditioned on $\mathcal{F}_t$. Now, the algorithm only fills the column corresponding to $(s,a)$ until the number of entries is smaller than $Np \leq \frac{N}{2}$. We conclude that:

$$|\{u : \hat{R}^{(t)}(h, (s,a)) = *\}| \geq N - Np \geq \frac{N}{2}.$$

This allows us to conclude $\mathbb{P}\left(\hat{R}^{(t)}_h(U_t, (s,a)) = * \Big| \mathcal{F}_t, (S^{(t)}_h, A^{(t)}_h) = (s,a)\right) \geq \frac{1}{2}$ and hence:

$$\begin{aligned} \mathbb{E}N_t &= \sum_{h=1}^H \mathbb{E}\mathbb{1}((S^{(t)}_h, A^{(t)}_h) \in \mathcal{G}^{(t)}_h) . \mathbb{1}(R^{(t)}_h(U_t, (S^{(t)}_h, A^{(t)}_h)) = *) \\ &\geq \frac{1}{2} \sum_{h=1}^H \mathbb{E}\mathbb{1}((S^{(t)}_h, A^{(t)}_h) \in \mathcal{G}^{(t)}_h) \\ &= \frac{1}{2} \sum_{h=1}^H P^{\hat{\Pi}^{\mathcal{G}}}_h(\mathcal{G}^{(t)}_h) \geq \frac{\epsilon}{8} \end{aligned} \tag{17}$$

In the last step we have used equation 15. The bound $|N_t| \leq H$ almost surely follows from definition. Now note that $\mathbb{E}[N_t^2 | \mathcal{F}_t] \leq H \mathbb{E}[N_t | \mathcal{F}_t]$.

$\square$

**Claim 6.** *For any $\tau \in \mathbb{N}$ and some $c_0 > 0$ small enough, we have:*

$$\mathbb{P}\left(\sum_{t=0}^{\tau-1} N_t < \frac{\epsilon \tau}{16}\right) \leq \exp(-c_0 \frac{\epsilon \tau}{H})$$

*Proof.* For $\frac{3}{4H} > \lambda > 0$, consider $M_t = -\frac{\lambda^2 \mathbb{E}[N_t^2|\mathcal{F}_t]}{1-\frac{\lambda H}{3}} + \lambda(\mathbb{E}[N_t|\mathcal{F}_t] - N_t)$. Now consider:

$$\mathbb{E}\exp(\sum_{t=0}^{\tau-1} M_t) = \mathbb{E}\left[\mathbb{E}[\exp(M_{\tau-1})|\mathcal{F}_{\tau-1}]\exp\left(\sum_{t=0}^{\tau-1} M_t\right)\right]$$

$$= \mathbb{E}[\exp(\lambda\mathbb{E}[N_{\tau-1}|\mathcal{F}_{\tau-1}] - \lambda N_t)|\mathcal{F}_{\tau-1}]\exp\left(\sum_{t=0}^{\tau-2} M_t\right)\exp\left(-\frac{\lambda^2\mathbb{E}[N_{\tau-1}^2|\mathcal{F}_{\tau-1}]}{1-\frac{\lambda H}{3}}\right)$$

$$\leq \mathbb{E}\exp\left(-\frac{\lambda^2\mathbb{E}[N_{\tau-1}^2|\mathcal{F}_{\tau-1}]}{1-\frac{\lambda H}{3}}\right)\exp\left(\sum_{t=0}^{\tau-2} M_t\right)\exp\left(-\frac{\lambda^2\mathbb{E}[N_{\tau-1}^2|\mathcal{F}_{\tau-1}]}{1-\frac{\lambda H}{3}}\right)$$

$$= \mathbb{E}\exp(\sum_{t=0}^{\tau-2} M_t) \tag{18}$$

In the first step we have used the fact that $\sum_{t=0}^{\tau-2} M_t$ is $\mathcal{F}_{\tau-1}$ measurable and the towering property of conditional expectation. In the third step, we have used the exponential moment bound given in Exercise 2.8.5 in Vershynin (2018), as applied to $N_\tau - \mathbb{E}[N_{\tau-1}|\mathcal{F}_{\tau-1}]$ along with the fact that $N_t \in [0, H]$ almost surely. From equation 18, we conclude that $\mathbb{E}\exp(\sum_{t=0}^{\tau} M_t) \leq 1$ and thus applying the Chernoff bound, we conclude that for any $\beta > 0$

$$\mathbb{P}\left(\sum_{t=0}^{\tau-1} -\frac{\lambda\mathbb{E}[N_t^2|\mathcal{F}_t]}{1-\frac{\lambda H}{3}} + (\mathbb{E}[N_t|\mathcal{F}_t] - N_t) > \beta\right) \leq \exp(-\lambda\beta)$$

Now, using item 2 from Claim 5, we conclude that

$$\mathbb{P}\left(\sum_{t=0}^{\tau-1} N_t < -\beta + \frac{3-4\lambda H}{3-\lambda H}\sum_{t=0}^{\tau-1}\mathbb{E}[N_t|\mathcal{F}_t]\right) \leq \exp(-\lambda\beta)$$

Now, using item 1 from Claim 5, we note that $\sum_{t=0}^{\tau-1}\mathbb{E}[N_t|\mathcal{F}_t] \geq \frac{\epsilon\tau}{8}$ almost surely. Setting $\lambda = \frac{1}{4H}$ and $\beta = c_0\epsilon\tau$ for some small enough constant $\epsilon$, we conclude:

$$\mathbb{P}\left(\sum_{t=0}^{\tau-1} N_t < \frac{\epsilon\tau}{16}\right) \leq \exp(-c_0\frac{\epsilon\tau}{H})$$

$\square$

Let $\tau^{\text{term}}$ denote the termination time for the algorithm. This is true since $\varphi(t)$ is increasing in $t$, $\varphi(t) \leq NpH|\mathcal{S}||\mathcal{A}|$, and strict inequality holds when $t < \tau^{\text{term}}$. For every $\tau < \tau^{\text{term}}$ we have $\varphi(\tau) = \sum_{t=0}^{\tau-1} N_\tau < NpH|\mathcal{S}||\mathcal{A}|$. Therefore, we have the following relationship between the events:

$$\{\tau^{\text{term}} > \tau\} \subseteq \{\sum_{t=1}^{\tau} N_\tau < Np|\mathcal{S}||\mathcal{A}|H\}$$

Setting $\tau = \frac{16Np|\mathcal{S}||\mathcal{A}|H}{\epsilon}$, we have:

$$\mathbb{P}(\tau^{\text{term}} > \tau) \leq \mathbb{P}\left(\sum_{t=1}^{\tau} N_\tau < Np|\mathcal{S}||\mathcal{A}|H\right) \leq \exp(-cNp|\mathcal{S}||\mathcal{A}|)$$

$\square$

### D.2 Proof of Lemma 8

*Proof.* By Remark 4.3 in Wagenmaker et al. (2022), we show that non-linear rewards can be handled by the reward free RL algorithm in Phase 1 as long all the reward are uniformly bounded in $[0, 1]$.

Let $B_{th}$ be the matrix $I + A_\phi$ in Algorithm 2 at step $t$ for horizon $h$. Similarly, let the corresponding projection $Q$ be $Q_{th}$. Recall that $Q_{th}$ is the projection onto an eigenspace of $B_{th}$. Now, suppose $S_{1:H}, A_{1:H} \sim \mathcal{M}_{U_t}(\hat{\Pi}^{Q_{t,h}})$ as in the algorithm. Let $\phi_{th} := \phi(S_h, A_h)$. Now, if $Q_{th} \neq 0$, then:

$$
\begin{aligned}
\phi_{th}^\mathsf{T} B_{th}^{-1} \phi_{th} &\geq \phi_{th}^\mathsf{T} Q_{th} B_{th}^{-1} Q_{th} \phi_{th} \\
&\geq \phi_{th}^\mathsf{T} Q_{th} \frac{I}{1 + \kappa^2} Q_{th} \phi_{th} \\
&= \frac{\|Q_{th}\phi_{th}\|^2}{1 + \kappa^2}
\end{aligned}
\tag{19}
$$

In the first step, we have used the fact that $Q_{th}$ is the projector to the eigenspace of $B_{th}^{-1}$. In the second step, we have used the fact that over the eigenspace corresponding to $Q_{th}$, the eigenvalues of $B_{th}^{-1}$ are at-least $\frac{1}{1+\kappa^2}$. We now invoke Assumption 2 in order to show that, along with the guarantees of reward free RL in phase 1, we conclude that:

$$
\mathbb{E}\left[\|Q_{th}\phi_{th}\|^2 \middle| Q_{th} \neq 0, B_{th}\right] \geq \gamma - \epsilon
\tag{20}
$$

Now, note by the fact that $Q_{th}$ is a projector and that $\|\phi_{th}\| \leq 1$, we have:

$$
\mathbb{E}\left[\|Q_{th}\phi_{th}\|^4 \middle| Q_{th} \neq 0, B_{th}\right] \leq \mathbb{E}\left[\|Q_{th}\phi_{th}\|^2 \middle| Q_{th} \neq 0, B_{th}\right]
\tag{21}
$$

Recall the Paley-Zygmund inequality which states that for any positive random variable $Z$, we must have: $\mathbb{P}(Z > \frac{\mathbb{E}Z}{2}) \geq \frac{1}{4}\frac{(\mathbb{E}Z)^2}{\mathbb{E}Z^2}$. Therefore,

$$
\begin{aligned}
\mathbb{P}\left[\phi_{th}^\mathsf{T} B_{th}^{-1} \phi_{th} > \frac{\gamma - \epsilon}{2(1 + \kappa^2)} \middle| Q_{th} \neq 0, B_{th}\right] &\geq \mathbb{P}\left[\|Q_{th}\phi_{th}\|^2 > \frac{\gamma - \epsilon}{2} \middle| Q_{th} \neq 0, B_{th}\right] \\
&\geq \mathbb{P}\left[\|Q_{th}\phi_{th}\|^2 > \frac{1}{2}\mathbb{E}\left[\|Q_{th}\phi_{th}\|^2 \middle| Q_{th} \neq 0, B_{th}\right] \middle| Q_{th} \neq 0, B_{th}\right] \\
&\geq \frac{1}{4}\frac{\mathbb{E}\left[\|Q_{th}\phi_{th}\|^2 \middle| Q_{th} \neq 0, B_{th}\right]^2}{\mathbb{E}\left[\|Q_{th}\phi_{th}\|^4 \middle| Q_{th} \neq 0, B_{th}\right]} \geq \frac{1}{4}\mathbb{E}\left[\|Q_{th}\phi_{th}\|^2 \middle| Q_{th} \neq 0, B_{th}\right] \\
&\geq \frac{\gamma - \epsilon}{4}
\end{aligned}
\tag{22}
$$

In the first step, we have used equation 19. In the second step, we have used equation 20. In the third step, we have used the Paley-Zygmund inequality and the moment bound in equation 21.

Define the stopping time $\tau = \inf\{t \leq T : Q_{th} = 0\}$ and $\tau = \infty$ if the set in the RHS is empty. Let $\Xi_t^0$ for $t \in \{0\} \cup \mathbb{N}$ be a sequence of i.i.d random variables with the law $\frac{\gamma - \epsilon}{2(1+\kappa^2)}\mathrm{Ber}(\frac{\gamma - \epsilon}{4})$. We consider the sequence of random variables $\Xi_t = \phi_{th}^\mathsf{T} B_{th}^{-1} \phi_{th}$ for $t < \tau$ and $\Xi_t = \Xi_t^0$ for $t \geq \tau$

Now, we apply the matrix determinant lemma which states that $\det(B + uu^\mathsf{T}) = \det(B)(1 + u^\mathsf{T} B^{-1} u)$. We note that $B_{(t+1)h} = B_{th} + \phi_{th}\phi_{th}^\mathsf{T}$. Therefore, whenever $t < \tau$, we must have:

$$
\det(B_{(t+1)h}) = \det(B_{th})(1 + \Xi_t)
\tag{23}
$$

Since $\|\phi_{th}\| \leq 1$ almost surely, we must have

$$
\mathrm{Tr}(B_{th}) = \sum_{i=1}^d \langle e_i, B_{th} e_i \rangle \leq d + t
$$

It is easy to show that for any PSD matrix, $A$, if $\mathrm{Tr}(A) \leq \alpha$, then $\det(A) \leq (\frac{\alpha}{d})^d$ (since trace is the sum of the eigenvalues and the determinant is the product). Combining the equations above, we conclude that whenever $t < \tau$, we must have:

$$
\left(\frac{t + 1 + d}{d}\right)^d \geq \prod_{s=0}^t (1 + \Xi_t)
$$

Therefore, the event

$$\{\tau > T\} \subseteq \{\left(\tfrac{T+1+d}{d}\right)^d \geq \prod_{s=0}^{T}(1 + \Xi_t)\} \tag{24}$$

**Claim 7.**

$$\mathbb{P}\left[\prod_{s=0}^{T}(1 + \Xi_t) \geq \left(1 + \frac{\gamma - \epsilon}{2(1 + \kappa^2)}\right)^{\frac{(\gamma - \epsilon)T}{8}}\right] \geq 1 - \exp\left(-c_0 T(\gamma - \epsilon)\right)$$

*Let $\kappa > 1$ and $T \geq C \frac{d\kappa^2}{(\gamma - \epsilon)^2} \log \frac{d\kappa}{\gamma - \epsilon}$, we have:*

$$\mathbb{P}\left[\prod_{s=0}^{T}(1 + \Xi_t) \geq \left(\tfrac{T+1+d}{d}\right)^d\right] \geq 1 - \exp\left(-c_0 T(\gamma - \epsilon)\right)$$

*Proof.* Let $N_T$ be the number of variables $(\Xi_t)_{t=0}^{T}$ such that $\Xi_t \geq \frac{\gamma - \epsilon}{2\kappa^2}$. Then, it is clear that $\prod_{s=0}^{T}(1 + \Xi_t) \geq (1 + \frac{\gamma - \epsilon}{2\kappa^2})^{N_T}$.

Therefore,

$$\mathbb{P}\left[\prod_{s=0}^{T}(1 + \Xi_t) \geq \left(1 + \frac{\gamma - \epsilon}{2(1 + \kappa^2)}\right)^{\frac{(\gamma - \epsilon)T}{8}}\right] \geq \mathbb{P}\left(N_T \geq \frac{(\gamma - \epsilon)T}{8}\right)$$

$$\geq \mathbb{P}\left(\mathrm{Bin}(T, \tfrac{\gamma - \epsilon}{4}) \geq \frac{(\gamma - \epsilon)T}{8}\right)$$

$$\geq 1 - \exp\left(-c_0 T(\gamma - \epsilon)\right) \tag{25}$$

Here Bin refers to the law of a binomial random variable. The first step follows from the fact that $\prod_{s=0}^{T}(1 + \Xi_t) \geq (1 + \frac{\gamma - \epsilon}{2(1 + \kappa^2)})^{N_T}$ almost surely. The second step follows from equation 22, which shows that conditioned on $Q_{th}, B_{th}$, the random variable $\mathbb{1}(\Xi_t \geq \frac{\gamma - \epsilon}{2(1 + \kappa^2)})$ stochastically dominates $\mathrm{Ber}(\frac{\gamma - \epsilon}{4})$. The last step follows from an application of Bernstein's inequality for binomial random variables. □

Now, using equation 24 along with Claim 7, we conclude:

$$\mathbb{P}(\tau > T) \leq \mathbb{P}\left(\left(\tfrac{T+1+d}{d}\right)^d \geq \prod_{s=0}^{T}(1 + \Xi_t)\right) \leq \exp(-c_0 T(\gamma - \epsilon))$$

□

### D.3 PROOF OF LEMMA 1

*Proof.* By the definition of Linear MDP, we must have $S_{h+1}|S_h, A_h \sim \sum_{i=1}^{d}\langle\phi(S_h, A_h), e_i\rangle\mu_{ih}(\cdot)$ and $A_{h+1}|S_{h+1} \sim \pi_{h+1}(\cdot|S_h)$. Therefore, for any bounded, measurable function $g : \mathcal{S} \times \mathcal{A} \to \mathbb{R}$, we must have:

$$\mathbb{E}g(S_{h+1}, A_{h+1}) = \mathbb{E}\left[\mathbb{E}\left[g(S_{h+1}, A_{h+1})\big|S_h, A_h\right]\right]$$

$$= \mathbb{E}\sum_{i=1}^{d}\langle\phi(S_h, A_h), e_i\rangle\int\mu_{i(h-1)}(ds)\pi_{h+1}(da|s)g(s, a)$$

$$= \sum_{i=1}^{d}\nu_{ih}\int\mu_{ih}(ds)\pi_{h+1}(da|s)g(s, a) \tag{26}$$

□

## D.4 Proof of Lemma 9

*Proof.* It is clear from the assumption that $\mathbb{E}\left[\int g(s_{(h+1)t}, a)\pi_{h+1}(da|s_t)|(\phi_{ht})_{t \leq T}\right] = \sum_{i=1}^{d}\langle\phi_{ht}, e_i\rangle \int \mu_{ih}(ds)\pi_{h+1}(da|s)g(s, a)$.

Note that

$$
\begin{aligned}
\sum_{t=1}^{T}\alpha_{ht,\nu}\phi_{ht} &= \sum_{t=1}^{T}\phi_{ht}^{\mathsf{T}}G_{\phi,h}^{-1}\nu\phi_{ht} \\
&= \sum_{t=1}^{T}\phi_{ht}\phi_{ht}^{\mathsf{T}}G_{\phi,h}^{-1}\nu = (\sum_{t=1}^{T}\phi_{ht}\phi_{ht}^{\mathsf{T}})G_{\phi,h}^{-1}\nu \\
&= G_{\phi,h}G_{\phi,h}^{-1}\nu = \nu
\end{aligned}
\tag{27}
$$

Therefore,

$$
\begin{aligned}
\mathbb{E}\left[\hat{\mathcal{T}}(g;\nu,\pi_h)|(\phi_l)_{t\in[T]}\right] &= \sum_{i=1}^{d}\langle\sum_{t=1}^{T}\alpha_{t,\nu}\phi_t, e_i\rangle\int\mu_{i(h-1)}(ds)\pi_h(da|s)g(s, a) \\
&= \sum_{i=1}^{d}\langle\nu, e_i\rangle\int\mu_{i(h-1)}(ds)\pi_h(da|s)g(s, a) = \mathcal{T}(g;\nu,\pi_h)
\end{aligned}
\tag{28}
$$

Note that, conditioned on $(\phi_t)_{t\in[T]}$, $\alpha_{ht,\nu}\int g(s_{(h+1)t}, a)\pi_{h+1}(da|s_{(h+1)t})$ are independent random variables bounded above by $\alpha_{ht,\nu}B$. Therefore, applying the Azuma-Hoeffding inequality, we conclude:

$$
\mathbb{P}\left(|\hat{\mathcal{T}}(g;\nu,\pi_h) - \mathcal{T}(g;\nu,\pi_h)| > \beta\bigg|(\phi_t)_{t\in[T]}\right) \leq 2\exp\left(-\frac{t^2}{2B^2\sum_t\alpha_{t,\nu}^2}\right)
$$

Now, observe that $\sum_t\alpha_{ht,\nu}^2 = \sum_t\nu^{\mathsf{T}}G_{\phi,h}^{-1}\nu \leq \frac{1}{\kappa^2}$ whenever $G_{\phi,h} \succeq \kappa^2 I$ This concludes the proof. □

## D.5 Proof of Lemma 10

*Proof.* Notice that:

$$
\begin{aligned}
\left|E_1^{\nu_1}(\Pi) - E_1^{\nu_1'}(\Pi')\right| &\leq \left|E_1^{\nu_1}(\Pi) - E_1^{\nu_1}(\Pi')\right| + \left|E_1^{\nu_1}(\Pi') - E_1^{\nu_1'}(\Pi')\right| \\
&\leq \left|E_1^{\nu_1}(\Pi) - E_1^{\nu_1}(\Pi')\right| + \|\nu_1 - \nu_1'\|_1 \\
&\leq \left\|\mathbb{E}\int\phi(S_1, a)\pi_1(da|S_1) - \mathbb{E}\int\phi(S_1, a)\pi_1'(da|S_1)\right\|_1 + \|\nu_1 - \nu_1'\|_1 \\
&\leq \sup_{(s,a)}\|\phi(s,a)\|_1\mathsf{TV}(\pi_1, \pi_1') + \|\nu_1 - \nu_1'\|_1 \leq \mathsf{TV}(\pi_1, \pi_1') + \|\nu_1 - \nu_1'\|_1
\end{aligned}
\tag{29}
$$

In the first, second and third steps we have used the triangle inequality. In the last step, we have used the fact that for any bounded function, and any probability measures $\mu, \nu$, we have $|\int f(x)\mu(dx) - \int f(x)\nu(dx)| \leq \sup_x|f(x)|\mathsf{TV}(\nu, \mu)$.

Now consider:

$$
\begin{aligned}
&\left|\|\mathcal{T}_j(\phi, \nu_{j-1}, \pi_j) - \nu_j\|_1 - \|\mathcal{T}_j(\phi, \nu_{j-1}', \pi_j') - \nu_j'\|_1\right| \\
&\leq \|\nu_j - \nu_j'\|_1 + \left\|\mathcal{T}_j(\phi, \nu_{j-1}, \pi_j) - \mathcal{T}_j(\phi, \nu_{j-1}', \pi_j')\right\|_1 \\
&\leq \|\nu_j - \nu_j'\|_1 + \left\|\mathcal{T}_j(\phi, \nu_{j-1}, \pi_j) - \mathcal{T}_j(\phi, \nu_{j-1}', \pi_j)\right\|_1 + \left\|\mathcal{T}_j(\phi, \nu_{j-1}', \pi_j) - \mathcal{T}_j(\phi, \nu_{j-1}', \pi_j')\right\|_1
\end{aligned}
\tag{30}
$$

Now, observe that:

$$\left\| \mathcal{T}_j(\phi, \nu_{j-1}, \pi_j) - \mathcal{T}_j(\phi, \nu'_{j-1}, \pi_j) \right\|_1 \leq \sum_{i=1}^{d} |\langle \nu_{j-1} - \nu'_{j-1}, e_i \rangle| \left\| \int \phi(s,a)\mu_{i(j-1)}(ds)\pi_j(da|s) \right\|_1$$

$$\leq \sum_{i=1}^{d} |\langle \nu_{j-1} - \nu'_{j-1}, e_i \rangle| = \|\nu_{j-1} - \nu'_{j-1}\|_1 \tag{31}$$

Where we recall $\sup_{i,h,\pi} \| \int \phi(s,a)\mu_{ih}(ds)\pi(da|s)\|_1 \leq 1$ as given in the definition of Linear MDP. Using the Hahn-Jordan decomposition of a signed measure, we conclude:

$$\left\| \mathcal{T}_j(\phi, \nu'_{j-1}, \pi_j) - \mathcal{T}_j(\phi, \nu'_{j-1}, \pi_j) \right\|_1$$

$$\leq \sum_{i=1}^{d} |\langle \nu'_{j-1}, e_i \rangle| \left\| \int \phi(s,a)\mu_{i(j-1)}(ds)(\pi_j(da|s) - \pi'_j(da|s)) \right\|_1$$

$$\leq \sum_{i=1}^{d} C_\mu |\langle \nu'_{j-1}, e_i \rangle| \mathsf{TV}(\pi_j, \pi'_j) \leq C_\mu \|\nu'_{j-1}\|_1 \mathsf{TV}(\pi_j, \pi'_j) \tag{32}$$

Combining equation 30, equation 31 and equation 32 we conclude:

$$\left| \|\mathcal{T}_j(\phi, \nu_{j-1}, \pi_j) - \nu_j\|_1 - \|\mathcal{T}_j(\phi, \nu'_{j-1}, \pi'_j) - \nu'_j\|_1 \right|$$

$$\leq \|\nu_j - \nu'_j\|_1 + C_\mu \|\nu'_{j-1}\|_1 \mathsf{TV}(\pi_j, \pi'_j) + \|\nu_{j-1} - \nu'_{j-1}\|_1 \tag{33}$$

Combining equation 29 and equation 33, we conclude the first inequality in the statement of the lemma.

With a reasoning very similar to that in equation 29, we have:

$$\left| \hat{E}_1^{\nu_1}(\Pi) - \hat{E}_1^{\nu'_1}(\Pi') \right| \leq \mathsf{TV}(\pi_1, \pi'_1) + \|\nu_1 - \nu'_1\|_1 \tag{34}$$

Using similar reasoning as in equation 33:

$$\left| \|\mathcal{T}_j(\phi, \nu_{j-1}, \pi_j) - \nu_j\|_1 - \|\mathcal{T}_j(\phi, \nu'_{j-1}, \pi'_j) - \nu'_j\|_1 \right|$$

$$\leq \|\nu_j - \nu'_j\|_1 + \left( \sum_{t=1}^{T} |(\nu_{j-1} - \nu'_{j-1})^\mathsf{T} G_{\phi,j-1}^{-1} \phi_{(j-1)t}| \right) + \left( \sum_{t=1}^{T} |(\nu'_{j-1})^\mathsf{T} G_{\phi,j-1}^{-1} \phi_{(j-1)t}| \right) \mathsf{TV}(\pi_j, \pi'_j) \tag{35}$$

Now note that for any $\nu \in \mathbb{R}^d$, we have:

$$\sum_{t=1}^{T} |\nu^\mathsf{T} G_{\phi,j-1}^{-1} \phi_{(j-1)t}| \leq \sqrt{T \sum_t |\nu^\mathsf{T} G_{\phi,j-1}^{-1} \phi_{(j-1)t}|^2}$$

$$= \sqrt{T \sum_{t=1}^{T} \nu^\mathsf{T} G_{\phi,j-1}^{-1} \phi_{(j-1)t} \phi_{(j-1)t}^\mathsf{T} G_{\phi,j-1}^{-1} \nu}$$

$$= \sqrt{T \nu^\mathsf{T} G_{\phi,j-1}^{-1} \nu}$$

$$\leq \sqrt{\tfrac{T}{\kappa^2}} \|\nu\|_2 \tag{36}$$

Here, in the first step we have used the fact that whenever $x \in \mathbb{R}^K$, we must have $\|x\|_1 \leq \sqrt{K}\|x\|_2$. In the third step, we have used the fact that $\sum_{t=1}^{T} \phi_{(j-1)t} \phi_{(j-1)t}^\mathsf{T} = G_{\phi,j-1}$ by definition. In the last step, we have used the fact that $\hat{G}_{\phi,j-1} \succeq \kappa^2 I$. Plugging this into equation 35, we conclude:

$$\left| \|\mathcal{T}_j(\phi, \nu_{j-1}, \pi_j) - \nu_j\|_1 - \|\mathcal{T}_j(\phi, \nu'_{j-1}, \pi'_j) - \nu'_j\|_1 \right|$$

$$\leq \|\nu_j - \nu'_j\|_1 + \sqrt{\tfrac{T}{\kappa^2}} \|\nu_{j-1} - \nu'_{j-1}\|_2 + \sqrt{\tfrac{T}{\kappa^2}} \|\nu'_{j-1}\|_2 \mathsf{TV}(\pi_j, \pi'_j) \tag{37}$$

Using this and the definition of $\hat{F}$ we conclude the second inequality in the statement of the lemma. equation 44 and equation 45 follow from a similar reasoning. □

## D.6 PROOF OF LEMMA 11

*Proof.* First consider the case $h = 1$. Let $g(s, a) := \phi(s, a)$. In this case, $\sup_{\nu \in \mathcal{B}_d(1)} |\hat{E}^{\nu,1}(\Pi) - E^{\nu,1}(\Pi)| \leq \|\mathcal{T}_0(g; \pi_1) - \hat{\mathcal{T}}_0(g; \pi_1)\|_1$. By equation 29 and equation 34, we conclude that $\pi_1 \to \mathcal{T}_0(\phi; \pi_1)$ and $\pi_1 \to \hat{\mathcal{T}}_0(\phi; \pi_1)$ are 1-Lipschitz with respect to $\mathsf{TV}()$ and $\|\cdot\|_1$.

$$\sup_{\Pi = \pi_1, \ldots, \pi_H \in \mathbb{Q}} \|\mathcal{T}_0(g; \pi_1) - \hat{\mathcal{T}}_0(g; \pi_1)\|_1 \leq \sup_{\Pi = \pi_1, \ldots, \pi_H \in \hat{\mathbb{Q}}_\eta} \|\mathcal{T}_0(g; \pi_1) - \hat{\mathcal{T}}_0(g; \pi_1)\|_1 + 2\eta$$

We apply Lemma 9 co-ordinate wise to the co-ordinates of $\phi$ and union bound over $\hat{\mathbb{Q}}_\eta$. We have:

$$\mathbb{P}\left(\sup_{\Pi = \pi_1, \ldots, \pi_H \in \mathbb{Q}} \|\mathcal{T}_0(g; \pi_1) - \hat{\mathcal{T}}_0(g; \pi_1)\|_1 > 2\eta + d\beta\right) \leq d|\hat{\mathbb{Q}}_\eta| \exp(-\frac{\beta^2 \kappa^2}{2}) \tag{38}$$

Now, consider $h > 1$. Consider any $\eta$ net over $\mathcal{B}_d(1)$, denoted by $\hat{\mathcal{B}}_{d,\eta}$ with respect to the norm $\|\cdot\|_1$. We can take $|\hat{\mathcal{B}}_{d,\eta}| \leq \exp(Cd \log(d/\eta))$ (Vershynin, 2018). Invoking Lemma 10, we conclude:

$$\sup_{\Pi \in \mathbb{Q}} \sup_{\nu \in \mathcal{B}_d(1)} |E^{\nu,h}(\Pi) - \hat{E}^{\nu,h}(\Pi)| \leq \sup_{\substack{\nu_1, \ldots, \nu_h \in \mathcal{B}_d(1) \\ \Pi \in \mathbb{Q}}} |\hat{F}(\Pi, \nu_1, \ldots, \nu_h) - F(\Pi, \nu_1, \ldots, \nu_h)|$$

$$\leq \sup_{\substack{\nu_1, \ldots, \nu_h \in \hat{\mathcal{B}}_{d,\eta} \\ \Pi \in \hat{\mathbb{Q}}_\eta}} |\hat{F}(\Pi, \nu_1, \ldots, \nu_h) - F(\Pi, \nu_1, \ldots, \nu_h)| + 2\left(1 + C_\mu + \sqrt{\frac{T}{\kappa^2}}\right)\eta h \tag{39}$$

Now, by the triangle inequality, we have:

$$|\hat{F}(\Pi, \nu_1, \ldots, \nu_h) - F(\Pi, \nu_1, \ldots, \nu_h)|$$
$$\leq \|\mathcal{T}_0(\phi, \pi_1) - \hat{\mathcal{T}}_0(\phi, \pi_1)\|_1 + \sum_{j=1}^{h-1} \|\mathcal{T}_j(\phi, \nu_j, \pi_{j+1}) - \hat{\mathcal{T}}_j(\phi, \nu_j, \pi_{j+1})\|_1 \tag{40}$$

Therefore, by invoking Lemma 9, along with union bound over every component in the sum in equation 40 and over the net in equation 39 we conclude that:

$$\mathbb{P}\left[\sup_{\substack{\nu_1, \ldots, \nu_h \in \hat{\mathcal{B}}_{d,\eta} \\ \Pi \in \hat{\mathbb{Q}}_\eta}} |\hat{F}(\Pi, \nu_1, \ldots, \nu_h) - F(\Pi, \nu_1, \ldots, \nu_h)| > \beta dh\right]$$
$$\leq 2dh|\hat{\mathbb{Q}}_\eta||\hat{\mathcal{B}}_{d,\eta}|^h \exp(-\frac{\beta^2 \kappa^2}{2}) \tag{41}$$

Combining equation 39 and equation 41, we conclude the second item in the statement of the lemma.

The concentration of $X_1$ and $X_h$ follow in a similar fashion, but here we consider an $\eta$ net even over $x$ and use the Lipschitzness results given in Lemma 10 and the fact that $x \to f(\phi; x)$ is 1 Lipschitz.

$\square$

## E PROOF OF THEOREM 3

**Lemma 9.** *Suppose $h \in [H-1]$, and $g : \mathcal{S} \times \mathcal{A} \to \mathbb{R}$ be such that $|g(s, a)| \leq B$ for every $(s, a)$. For any policy $\pi_h$ and any $\nu$ such that $\|\nu\|_2 \leq 1$, we must have:*

$$\mathbb{P}\left(|\hat{\mathcal{T}}_h(g; \nu, \pi_h) - \mathcal{T}_h(g; \nu, \pi_h)| > \beta \,\Big|\, (\phi_{ht})_{t \in [T]}, G_{\phi,h} \succeq \kappa^2 I\right) \leq 2\exp\left(-\frac{\beta^2 \kappa^2}{2B^2}\right)$$

**Lemma 10.** *Let* $\Pi = (\pi_1, \ldots, \pi_H)$, $\Pi' = (\pi'_1, \ldots, \pi'_H)$ *be policies in* $\mathbb{Q}$. *Conditioned on the event* $G_{\phi,h} \succeq \kappa^2 I$, *the following hold:*

$$|F(\Pi, \nu_1, \ldots, \nu_{h-1}, \nu_h) - F(\Pi', \nu'_1, \ldots, \nu'_{h-1}, \nu'_h)|$$

$$\leq \left( \sum_{j=2}^{h} C_\mu \mathsf{TV}(\pi_j, \pi'_j) \|\nu_j\|_1 + 2\|\nu_j - \nu'_j\|_1 \right) + \mathsf{TV}(\pi_1, \pi'_1) + 2\|\nu_1 - \nu'_1\|_1 \quad (42)$$

$$|\hat{F}(\Pi, \nu_1, \ldots, \nu_{h-1}, \nu_h) - \hat{F}(\Pi', \nu'_1, \ldots, \nu'_{h-1}, \nu'_h)|$$

$$\leq \sqrt{\frac{T}{\kappa^2}} \left( \sum_{j=2}^{h} \mathsf{TV}(\pi_j, \pi'_j) \|\nu_j\|_2 + \|\nu_j - \nu'_j\|_2 \right) + \mathsf{TV}(\pi_1, \pi'_1) + \sum_{j=1}^{h} \|\nu_j - \nu'_j\|_1 \quad (43)$$

*Suppose* $x \in \mathcal{S}^{d-1}$

$$|\mathcal{T}_h(f(\cdot; x), \nu, \pi_h) - \mathcal{T}_h(f(\cdot; x'), \nu', \pi'_h)|$$

$$\leq 2C_\mu(\sqrt{d} + \xi d) \left( \|\nu - \nu'\|_1 + \mathsf{TV}(\pi_h, \pi'_h) + \|x - x'\|_2 \|\nu\|_1 \right) \quad (44)$$

$$|\hat{\mathcal{T}}_h(f(\cdot; x), \nu, \pi_h) - \hat{\mathcal{T}}_h(f(\cdot; x'), \nu', \pi'_h)|$$

$$\leq 2\sqrt{\frac{T}{\kappa^2}}(\sqrt{d} + \xi d) \left( \|\nu - \nu'\|_2 + \mathsf{TV}(\pi_h, \pi'_h) + \|x - x'\|_2 \|\nu\|_2 \right) \quad (45)$$

**Lemma 11.** *Condition on the event* $G_{\phi,h} \succeq \kappa^2 I$ *for every* $h \in [H]$. *Fix some* $\eta > 0$ *and let* $\hat{\mathbb{Q}}_\eta$ *denote any* $\eta$-*net over* $\mathbb{Q}$. *With probability at-least* $1 - \delta/4$, *the following hold simultaneously:*

1.

$$\sup_\nu \sup_{\Pi \in \mathbb{Q}} |E_1^\nu(\Pi) - \hat{E}_1^\nu(\Pi)| \leq C \frac{d}{\kappa} \sqrt{\log\left( \frac{d|\hat{\mathbb{Q}}_\eta|}{\delta} \right)} + C\eta.$$

2. *For* $h > 1$:

$$\sup_{\Pi \in \mathbb{Q}} \sup_{\nu \in \mathcal{B}_d(1)} |E_h^\nu(\Pi) - \hat{E}_h^\nu(\Pi)| \leq \frac{CdH}{\kappa} \sqrt{\log\left( \frac{dH|\hat{\mathbb{Q}}_\eta|}{\delta} \right) + Hd \log\left( \frac{d}{\eta} \right)} + C\left( \sqrt{\frac{T}{\kappa^2}} \right) \eta H$$

3. $X_1 := \sup_{\Pi = (\pi_1, \ldots, \pi_H) \in \mathbb{Q}} \left| \inf_{x \in \mathcal{S}^{d-1}} \hat{\mathcal{T}}_1(f(\cdot; x), \pi_1) - \inf_{x \in \mathcal{S}^{d-1}} \mathcal{T}_1(f(\cdot; x), \pi_1) \right|$

$$X_1 \leq \frac{C(\sqrt{d} + \xi d)}{\kappa} \sqrt{\log\left( \frac{|\hat{\mathbb{Q}}_\eta|}{\delta} \right) + d \log\left( \frac{d}{\eta} \right)} + C\eta(\sqrt{d} + \xi d)$$

4. $X_h := \sup_{\substack{\nu \in \mathcal{B}(1) \\ \Pi = (\pi_1, \ldots, \pi_H) \in \mathbb{Q}}} \left| \inf_{x \in \mathcal{S}^{d-1}} \hat{\mathcal{T}}_h(f(\cdot; x); \nu, \pi_h) - \inf_{x \in \mathcal{S}^{d-1}} \mathcal{T}_h(f(\cdot; x); \nu, \pi_h) \right|$

$$X_h \leq \frac{C(\sqrt{d} + \xi d)}{\kappa} \sqrt{\log\left( \frac{|\hat{\mathbb{Q}}_\eta| H}{\delta} \right) + d \log\left( \frac{d}{\eta} \right)} + C\eta(\sqrt{d} + \xi d)\left( C_\mu + \sqrt{\frac{T}{\kappa^2}} \right)$$

**Lemma 12.** $\Pi = (\pi_1, \ldots, \pi_H)$. *For any* $\eta \geq 0$, *and* $h \in [H]$, *suppose* $E_h^\nu(\Pi) \leq \eta$. *Then, we have:*

$$\|\mathbb{E}\phi(S_h, A_h) - \nu\|_1 \leq \eta.$$

*Proof.* Let $S_{1:H}, A_{1:H} \sim \mathcal{M}(\Pi)$. By Lemma 1, we conclude that: $\mathbb{E}\phi(S_1, A_1) = \mathcal{T}_1(\phi, \pi_1)$. Therefore we conclude the lemma for the case $h = 1$. Now let $h > 1$.

Now, note that for $j > 1$, we have: $\mathbb{E}\phi(S_j, A_j) = \mathcal{T}_j(\phi, \mathbb{E}\phi(S_{j-1}, A_{j-1}), \pi_j)$. There exists a sequence $\nu_1, \ldots, \nu_{h-1}$ such that

$$E_1^{\nu_1}(\Pi) + \sum_{j=2}^{h} \|\mathcal{T}_j(\phi, \nu_{j-1}, \pi_j) - \nu_j\|_1 \leq \eta_0$$

Letting $E_1^{\nu_1}(\Pi) =: \eta_1$, $\|\mathcal{T}_j(\phi, \nu_{j-1}, \pi_j) - \nu_j\|_1 =: \eta_j$, we have from the case $h = 1 : \|\mathbb{E}\phi(S_1, A_1) - \nu_1\|_1 \leq \eta_1$.

$$
\begin{aligned}
\|\mathbb{E}\phi(S_j, A_j) - \nu_j\|_1 &= \|\mathcal{T}_j(\phi, \mathbb{E}\phi(S_{j-1}, A_{j-1}), \pi_j) - \nu_j\|_1 \\
&\leq \|\mathcal{T}_j(\phi, \mathbb{E}\phi(S_{j-1}, A_{j-1}), \pi_j) - \mathcal{T}_j(\phi, \nu_{j-1}, \pi_j)\|_1 + \|\mathcal{T}_j(\phi, \nu_{j-1}, \pi_j) - \nu_j\|_1 \\
&\leq \|\nu_{j-1} - \mathbb{E}\phi(S_{j-1}, A_{j-1})\|_1 + \eta_j
\end{aligned}
\tag{46}
$$

We have used equation 31 in the last step. Continuing recursively, we conclude the result $\qquad\square$

*Proof of Theorem 3.* We condition on the event described in Lemma 11. We suppose that $\kappa$, $\eta$ and $\eta_0$ are related as in the statement of the theorem. We will apply these values whenever we invoke the concentration bounds obtained from Lemma 11 in all the inequalities below. First consider $h = 1$. Let $\hat{\Pi}^{f,1} = (\pi_H^{f,1}, \ldots, \pi_H^{f,H})$. By item 3 in Lemma 11, we have (with $X_1$ as defined in the lemma):

$$
\inf_{x \in \mathcal{S}^{d-1}} \hat{\mathcal{T}}_1(f(;x), \pi_1^{f,1}) \geq \sup_{\Pi \in \mathbb{Q}} \inf_{x \in \mathcal{S}^{d-1}} \mathcal{T}_1(f(;x), \pi_1) - X_1 \geq \zeta - X_1 \geq \tfrac{3\zeta}{4}
$$

Similarly, we have:

$$
\inf_{x \in \mathcal{S}^{d-1}} \mathcal{T}_1(f(;x), \pi_1^{f,1}) \geq \inf_{x \in \mathcal{S}^{d-1}} \hat{\mathcal{T}}_1(f(;x), \pi_1^{f,1}) - \tfrac{\zeta}{4}
$$

Combining the two displays above, we conclude the theorem for $h = 1$. Now consider $h > 1$. We will first show that the constraint $\hat{E}^{\nu, h-1}(\Pi) \leq \eta_0$ is feasible for some $\Pi \in \mathbb{Q}$ and some $\nu$. Note that, for any policy $\Pi$ there exists a $\nu_1, \ldots, \nu_{h-1} \in \mathbb{R}^d$ such that $\mathbb{E}\phi(S_j, A_j) = \nu_j$ whenever $S_{1:H}, A_{1:H} \sim \mathcal{M}(\Pi)$. For the choice $\nu = \nu_{h-1}$, we must have $E^{\nu, h-1}(\Pi) = 0$. Now, by item 1 and 2 of Lemma 11, we conclude that $\hat{E}^{\nu, h-1} \leq \eta_0$. Therefore this optimization is feasible.

Consider the solutions to the optimization problem given by $\hat{\nu}$ and $\hat{\Pi}^{f,h}$. Note again from Lemma 11 that $E^{\hat{\nu}, h-1}(\hat{\Pi}^{f,h}) \leq \hat{E}^{\hat{\nu}, h-1}(\hat{\Pi}^{f,h}) + \eta_0 \leq 2\eta_0$. Now, applying Lemma 12, we conclude that whenever $S_{1:H}, A_{1:H} \sim \mathcal{M}(\hat{\Pi}^{f,h})$

$$
\|\mathbb{E}\phi(S_{h-1}, A_{h-1}) - \hat{\nu}\|_1 \leq 2\eta_0
$$

By a similar reasoning as the case $h = 1$, we conclude:

$$
\inf_{x \in \mathcal{S}^{d-1}} \mathcal{T}_h(f(;x), \hat{\nu}, \pi_h^{f,h}) \geq 3\tfrac{\zeta}{4}
$$

Now, applying equation 44, we conclude:

$$
\begin{aligned}
\inf_{x \in \mathcal{S}^{d-1}} \mathbb{E}f(S_h, A_h; x) &= \inf_{x \in \mathcal{S}^{d-1}} \mathcal{T}_h(f(;x), \mathbb{E}\phi(S_{h-1}, A_{h-1}), \pi_h^{f,h}) \\
&\geq \inf_{x \in \mathcal{S}^{d-1}} \mathcal{T}_h(f(;x), \hat{\nu}, \pi_h^{f,h}) - \sup_{x \in \mathcal{S}^{d-1}} |\mathcal{T}_h(f(;x), \mathbb{E}\phi(S_{h-1}, A_{h-1}), \pi_h^{f,h}) - \mathcal{T}_h(f(;x), \hat{\nu}, \pi_h^{f,h})| \\
&\geq \frac{3\zeta}{4} - 2C_\mu(\sqrt{d} + \xi d) \left(\|\hat{\nu} - \mathbb{E}\phi(S_{h-1}, A_{h-1})\|_1\right) \geq \frac{\zeta}{2}
\end{aligned}
\tag{47}
$$

In the last step, we have used the lipschitzness bound for $\mathcal{T}_h$ given in Lemma 10. We will show that the conditions given in equation 2 are satisfied for $\psi(S_h, A_h)$ with parameters $\zeta/2$ instead of $\zeta$.

$\|\psi(S_h, A_h)\|_2 \leq 1$ almost surely follows from the definition of $\psi$. Now, $\mathbb{E}f(S_h, A_h, x) \geq \frac{\zeta}{2}$ for every $x \in \mathcal{S}^{d-1}$ implies $\mathbb{E}|\langle x, \psi(S_h, A_h)\rangle| \geq \frac{\zeta}{2\sqrt{d}}$. Using the definition of $f(S_h, A_h, x)$ (see Section 5) and the fact that $\mathbb{E}f(S_h, A_h, x) \geq \frac{\zeta}{2}$ as established above, we conclude that for every $x \in \mathcal{S}^{d-1}$, we also have:

$$
\begin{aligned}
d\xi \mathbb{E}\langle x, \psi(S_h, A_h)\rangle^2 &\leq \sqrt{d}\mathbb{E}|\langle x, \psi(S_h, A_h)\rangle| - \frac{\zeta}{2} \\
&\leq \sqrt{d}\mathbb{E}|\langle x, \psi(S_h, A_h)\rangle| \leq \sqrt{d}\sqrt{\mathbb{E}\langle x, \psi(S_h, A_h)\rangle^2}
\end{aligned}
\tag{48}
$$

In the second step, we have used Jensen's inequality. From this, we conclude $\mathbb{E}\langle x, \psi(S_h, A_h)\rangle^2 \leq \frac{1}{d\xi^2}$ for every $x \in \mathcal{S}^{d-1}$ and thence $\mathbb{E}\psi(S_h, A_h)\psi(S_h, A_h)^\intercal \preceq \frac{1}{d\xi^2}$. $\qquad\square$

# F    PROOF OF THEOREM 4

Let the unknown row set in the iteration $t$ in the matrix estimation procedure of Section 6.1 be denoted by $\bar{I}_{t-1}$. For the analysis, we will use the convention that $\bar{I}_t = \emptyset$ if the procedure terminates before the $t$-th iteration. Suppose $K_t$ is such that for every $t \leq \log N$, we have: $K_t |\bar{I}_{t-1}| \geq C \frac{r|\bar{I}_{t-1}| + dr}{\zeta^2 \xi^2} \log \frac{d}{\zeta \xi} + C \frac{\log\left(\frac{\log N}{\delta}\right)}{\zeta^2 \xi^2}$. We will then show that the event $\{|\bar{I}_t| \leq \frac{1}{10} |\bar{I}_{t-1}| \forall t \leq \log N\} \cap \{\hat{\Theta}_i = \Theta_i^*, \forall i \in \bar{I}_{\log N}^{\complement}\}$ has probability at-least $1 - \delta$. To show this, it is sufficient to consider the step $t = 1$ with $\bar{I}_0 = [N]$, $K_1 = K$, $\Psi^{(1)} = \Psi$ and show that with probability $1 - \frac{\delta}{\log N}$, $\bar{I}_1 \leq \frac{9N}{10}$ and $\hat{\Theta}_i = \Theta_i^*$ for every $i \in \bar{I}_1^{\complement}$. The result then follows from a union bound. We will therefore establish the following structural lemma and prove the Theorem 4. The rest of the section is then dedicated to proving Lemma 13.

**Lemma 13.** *Suppose the distribution of $\psi_{ik}$ satisfies equation 2. Let $K \geq \frac{C(r + \frac{dr}{N})}{\zeta^2 \xi^2} \log \frac{d}{\zeta \xi}$. Let $\mathcal{Y}(\Psi)$ denote the set of all matrices $\Delta$ with rank at most $2r$ such that $L(\Delta, \Psi) = 0$. Let $I_{\mathcal{Z}}(\Delta) = \{i \in [N] : \Delta_i \neq 0\}$. With probability $1 - \exp(-c\zeta^2 \xi^2 NK)$ we must have:*

$$\mathcal{Y}(\Psi) \cap \left\{\Delta : |I_{\mathcal{Z}}(\Delta)| > \tfrac{N}{10}\right\} = \emptyset$$

*Proof of Theorem 4.* Let $\bar{\Theta}$ be the rank $\leq r$ matrix found satisfying $L(\bar{\Theta} - \Theta^*, \Psi^{(t)}) = 0$. By Lemma 13, we have that $|I_{\mathcal{Z}}(\bar{\Theta} - \Theta^*)| \leq \frac{N}{10}$ with probability at-least $1 - \frac{\delta}{\log N}$ (by setting $K = K_1$ as in the statement of Theorem 4). By Lemma 14, the probability that there exists an $i \in [N]$ such that $\bar{\Theta}_i \neq \Theta_i^*$, and $\sum_{k=1}^K \left|\langle \bar{\Theta}_i, \tilde{\psi}_{ik}\rangle - \theta_{ik}^*\right|^2 = 0$ is at most $|I| \cdot \exp(-c\zeta^2 \xi^2 K) \leq \delta \cdot N^{-c}$ for some large constant $c$. From this we conclude that $\bar{\Theta}_i = \Theta_i^*$ for every $i \in \bar{I}_1^{\complement}$. □

**Lemma 14.** *Fix any $\hat{\Theta}$. Suppose the distribution of $(\psi_{ik})_{i\in[N], k\in[K]}$ satisfies equation 2. Then, there exists a small enough constant $c$ such that:*

$$\mathbb{P}\left(\exists i \text{ s.t. } \sum_{k=1}^K \left|\langle \hat{\Theta}_i, \psi_{ik}\rangle - \theta_{ik}^*\right|^2 < \frac{K\zeta^4 \xi^2 \|\hat{\Theta}_i - \Theta_i^*\|^2}{32d}\right) \leq |I| \cdot \exp(-c\zeta^2 \xi^2 K).$$

*Proof.* Consider the Paley-Zygmund inequality, which states that for any positive random variable $Z$,

$$\mathbb{P}\left(Z \geq \tfrac{\mathbb{E}Z}{2}\right) \geq \frac{(\mathbb{E}Z)^2}{4\mathbb{E}Z^2}.$$

Suppose $i \in I$ and denote $\Gamma_i := \hat{\Theta}_i - \Theta_i^*$. By the properties of $\psi_{ik}$, we have that $\mathbb{E}|\langle \Gamma_i, \psi_{ik}\rangle| \geq \frac{\zeta \|\Gamma_i\|}{\sqrt{d}}$ and $\mathbb{E}|\langle \Gamma_i, \psi_{ik}\rangle|^2 \leq \frac{\|\Gamma_i\|^2}{\xi^2 d}$.

Applying the Paley-Zygmund inequality to the random variable $|\langle \Gamma_i, \psi_{ik}\rangle|$, we conclude the result in equation 52:

$$\mathbb{P}\left(|\langle \psi_{ik}, \Gamma_i\rangle| \geq \frac{\zeta \|\Gamma_i\|}{2\sqrt{d}}\right) \geq \frac{\zeta^2 \xi^2}{4} \tag{49}$$

Let $p_0 := \frac{\zeta^2 \xi^2}{4}$. Let $N(\Gamma_i, \Psi_i) := \sum_{k=1}^K \mathbb{1}\left(|\langle \psi_{ik}, \Gamma_i\rangle| > \frac{\zeta}{2\sqrt{d}}\right)$. Clearly, $\sum_{k=1}^K \left|\langle \hat{\Theta}_i, \psi_{ik}\rangle - \theta_{ik}^*\right|^2 \geq \frac{\zeta^2 \|\Gamma_i\|^2}{4d} \cdot N(\Gamma_i, \Psi_i)$. Therefore, we have:

$$\mathbb{P}\left(\sum_{k=1}^K \left|\langle \hat{\Theta}_i, \psi_{ik}\rangle - \theta_{ik}^*\right|^2 < \frac{K\zeta^4 \xi^2 \|\Gamma_i\|^2}{32d}\right) \leq \mathbb{P}\left(N(\Gamma_i, \Psi_i) < \frac{K\zeta^2 \xi^2}{8}\right)$$

$$\leq \mathbb{P}\left(\mathrm{Bin}(K, p_0) \leq \frac{Kp_0}{2}\right)$$

$$\leq \exp(-cp_0 K) \tag{50}$$

Here $\text{Bin}(K, p_0)$ denotes the binomial random variable. In the second step we have used the fact that $N(\Gamma_i, \Psi_i)$ is a sum of $K$ independent Bernoulli random variables with probability of being 1 for each of them being at-least $p_0 = \frac{\zeta^2 \xi^2}{4}$. In the last step, we have used Sanov's theorem for large deviations. In the last step we have used Bernstein's inequality for concentration of sums of Bernoulli random variables (see Boucheron et al. (2013)). The statement of the result then follows from a union bound argument over $i \in I$. $\qquad\square$

### F.1 PROOF OF LEMMA 13

Suppose $I \neq \emptyset$, $I \subseteq [N]$ be any fixed subset. By $\mathcal{M}(N, d, I, 2r)$, we denote the set of all $N \times d$ matrices $\Delta$ with rank at-most $2r$ such that $\|\Delta_i\| > 0$ for all $i \in I$. By $\mathcal{B}(N, d, I, 2r)$ we denote the set of all $N \times d$ matrices with rank at-most $2r$ such that $\|\Gamma_i\| = 1$ whenever $i \in I$.

**Lemma 15.** *Suppose* $\inf_{\Gamma \in \mathcal{B}(N,d,I,2r)} L(\Gamma, \Psi) > 0$. *Then,* $L(\Delta, \Psi) > 0$ *for every* $\Delta \in \mathcal{M}(N, d, I, 2r)$

*Proof.* For every $\Delta \in \mathcal{M}(N, d, I, 2r)$, we construct $\Gamma$ such that:

$$\Gamma_i = \begin{cases} \frac{\Delta_i}{\|\Delta_i\|} & \text{whenever } i \in I \\ 0 & \text{otherwise} \end{cases} \tag{51}$$

Now, by hypothesis, $L(\Gamma, \Psi) > 0$. This implies, there exists an $i \in I$ and $k \in K$ such that $|\langle \psi_{ik}, \Gamma_i \rangle| > 0$. This implies $|\langle \psi_{ik}, \Delta_i \rangle| > 0$ and thence we conclude that $L(\Delta, \Psi) > 0$. $\qquad\square$

**Lemma 16.** *Suppose* $\Gamma$ *is such that* $\|\Gamma_i\| = 1$ *for every* $i \in I$. *Suppose the distribution of* $(\psi_{ik})_{i \in [N], k \in [K]}$ *satisfy equation 2. Then, there exists a small enough constant $c$ such that:*

$$\mathbb{P}\left(L(\Gamma, \Psi) < \frac{|I|\zeta^4 \xi^2}{32 N d}\right) \leq |I|^2 K^2 \exp(-c\zeta^2 \xi^2 |I| K)$$

*Proof.* Consider the Paley-Zygmund inequality, which states that for any positive random variable $Z$,

$$\mathbb{P}\left(Z \geq \frac{\mathbb{E}Z}{2}\right) \geq \frac{(\mathbb{E}Z)^2}{4\mathbb{E}Z^2}.$$

Suppose $i \in I$. By the properties of $\psi_{ik}$, we have that $\mathbb{E}|\langle \Gamma_i, \psi_{ik} \rangle| \geq \frac{\zeta}{\sqrt{d}}$ and $\mathbb{E}|\langle \Gamma_i, \psi_{ik} \rangle|^2 \leq \frac{1}{\xi^2 d}$

Applying the Paley-Zygmund inequality to the random variable $|\langle \Gamma_i, \psi_{ik} \rangle|$, we conclude the result in equation 52:

$$\mathbb{P}\left(|\langle \psi_{ik}, \Gamma_i \rangle| \geq \frac{\zeta}{2\sqrt{d}}\right) \geq \frac{\zeta^2 \xi^2}{4} \tag{52}$$

Let $p_0 := \frac{\zeta^2 \xi^2}{4}$. Let $N(\Gamma, \Psi) := \sum_{i \in I} \sum_{k=1}^{K} \mathbb{1}\left(|\langle \psi_{ik}, \Gamma_i \rangle| > \frac{\zeta}{2\sqrt{d}}\right)$. Clearly, $L(\Gamma, \Psi) \geq \frac{\zeta^4}{4dNK} N(\Gamma, \Psi)$ almost surely. Therefore, we have:

$$\begin{aligned} \mathbb{P}\left(L(\Gamma, \Psi) < \frac{|I|\zeta^4 \xi^2}{32 N \sqrt{d}}\right) &\leq \mathbb{P}\left(N(\Gamma, \Psi) < \frac{|I|K \zeta^2 \xi^2}{8}\right) \\ &\leq \mathbb{P}\left(\text{Bin}(|I|K, p_0) \leq \frac{|I|K p_0}{2}\right) \\ &\leq \exp(-cp_0 |I| K) \end{aligned} \tag{53}$$

Here $\text{Bin}(|I|K, p_0)$ denotes the binomial random variable. In the second step we have used the fact that $N(\Gamma, \Psi)$ is a sum of $|I|K$ independent Bernoulli random variables with probability of being 1 for each of them being at-least $p_0 = \frac{\zeta^2 \xi^2}{4}$. In the last step, we have used Sanov's theorem for large deviations. In the last step we have used Bernstein's inequality for concentration of sums of Bernoulli random variables (see Boucheron et al. (2013)) $\qquad\square$

**Lemma 17.** *Suppose the distribution of $(\psi_{ik})_{i\in[N],k\in[K]}$ satisfy equation 2. Let $|I| \geq \frac{N}{10}$. There exist positive constants $c_0, c, C$ such that whenever $KN \geq \frac{Cr(N+d)}{\zeta^2\xi^2} \log \frac{d}{\zeta\xi}$, we have:*

$$\mathbb{P}\left(\inf_{\Gamma\in\mathcal{B}(N,d,I,2r)} L(\Gamma,\Psi) < c_0\frac{\zeta^4\xi^2}{d}\right) \leq \exp(-c\zeta^2\xi^2NK)$$

*Proof.* It is sufficient to prove this result for $\Gamma \in \mathcal{B}_0(N,d,I,2r) \subseteq \mathcal{B}(N,d,I,2r)$, which is the set of all matrices such that $\|\Gamma_i\| = 1$ for every $i \in I$ and 0 otherwise. Define $\|\Gamma\|_{1,2,\mathsf{T}} := \frac{1}{N}\sum_{i=1}^N \|\Gamma_i\|$. Suppose $\hat{\Gamma} \in \mathcal{B}_0(N,d,I,2r)$ is such that $\|\Gamma - \hat{\Gamma}\|_{1,2,\mathsf{T}} < \eta$. Then,

$$L(\Gamma,\Psi) = \frac{1}{NK}\sum_{i=1}^N\sum_{k=1}^N |\langle\Gamma_i,\psi_{ik}\rangle|^2$$

$$\geq \frac{1}{NK}\sum_{i=1}^N\sum_{k=1}^N |\langle\hat{\Gamma}_i,\psi_{ik}\rangle|^2 - 2|\langle\hat{\Gamma}_i - \Gamma_i,\psi_{ik}\rangle||\langle\hat{\Gamma}_i,\psi_{ik}\rangle| \tag{54}$$

$$= L(\hat{\Gamma},\Psi) - \|\Gamma - \hat{\Gamma}\|_{1,2,\mathsf{T}} \geq L(\hat{\Gamma},\Psi) - 2\eta \tag{55}$$

In the third step, we have used the fact that $\|\psi_{ik}\| \leq 1$ and the Cauchy-Schwarz inequality to imply $|\langle\hat{\Gamma}_i - \Gamma_i,\psi_{ik}\rangle| \leq \|\hat{\Gamma}_i - \Gamma_i\|$. Therefore, given any $\eta$ net of $\mathcal{B}_0(N,d,I,2r)$, denoted by $\hat{\mathcal{B}}_{0,\eta}$, we must have:

$$\inf_{\Gamma\in\mathcal{B}_0(N,d,I,2r)} L(\Gamma,\Psi) \geq \inf_{\Gamma\in\hat{\mathcal{B}}_{0,\eta}} L(\hat{\Gamma},\Psi) - 2\eta \tag{56}$$

We will now parametrize $\mathcal{B}_0(N,d,I,2r)$ as follows:

**Claim 8.** *Every $\Gamma \in \mathcal{B}_0(N,d,I,2r)$ can be written as*

$$\Gamma_i = \begin{cases} \sum_{k=1}^{2r} u_{ik}v_k \text{ if } i \in I \\ 0 \text{ otherwise} \end{cases} \tag{57}$$

*Where $v_1,\ldots,v_{2r}$ are orthonormal vectors in $\mathbb{R}^d$ and $u_i = (u_{ik})_{k=1}^{2r} \in \mathbb{R}^{2r}$ are such that $\|u_i\| = 1$.*

*Proof.* By the singular value decomposition, we have: $\Gamma = W\Sigma V^\mathsf{T}$ for orthogonal matrices $W, V$ and the singular value matrix $\Sigma$. Therefore, $\Gamma_{ij} = \sum_{k=1}^{2r} w_{ik}\sigma_k v_{kj}$ Denoting $u_{ik} := w_{ik}\sigma_k$, we note that $\Gamma_i = \sum_{k=1}^{2r} u_{ik}v_k$, where $v_k$ is the $k$-th column of $V$.

Now, it remains to show that $\|u_i\| = 1$. By ortho-normality of $v_1,\ldots,v_{2r}$ and the definition of $\Gamma$, we have: $1 = \|\Gamma_i\|^2 = \sum_{k=1}^{2r} |u_{ik}|^2 = \|u_i\|^2$ $\square$

Therefore, we construct an $\eta$-net for $\mathcal{B}_0(N,d,I,2r)$ as follows: consider any $\eta/2$-net over the sphere $\mathcal{S}^{2r-1}$, denoted by $\hat{\mathcal{S}}_{\frac{\eta}{2}}(2r)$ with respect to the Euclidean norm. Similarly, consider any $\frac{\eta}{2\sqrt{2r}}$-net over the sphere $\mathcal{S}^{d-1}$, denoted by $\hat{\mathcal{S}}_{\frac{\eta}{2\sqrt{2r}}}(d)$. We draw $(u_i)_{i\in I}, (v_k)_{k\in[2r]}$ from the set $\prod_{i\in I}\hat{\mathcal{S}}_{\frac{\eta}{2}}(2r)\prod_{k\in[2r]}\hat{\mathcal{S}}_{\frac{\eta}{2\sqrt{2r}}}(d)$ and take $\hat{\mathcal{B}}_{0,\eta}$ to be the set of all $\hat{\Gamma}(u,v)$ of the form given in Claim 8.

**Claim 9.** *$\hat{\mathcal{B}}_{0,\eta}$ is an $\eta$ net for $\mathcal{B}_0(N,d,I,2r)$ with respect to the norm $\|\cdot\|_{1,2,\mathsf{T}}$.*

$$|\hat{\mathcal{B}}_{0,\eta}| \leq \exp\left(2dr\log(\frac{4\sqrt{2r}}{\eta}+1) + 2|I|r\log(\frac{4}{\eta}+1)\right)$$

*Proof of Claim 9.* Let $\Gamma \in \mathcal{B}_0(N,d,I,2r)$. Let $(u_i),(v_k)$ be such that: Claim 8, $\Gamma_i = \sum_{k=1}^{2r} u_{ik}v_k$. By construction, there exists $\hat{\Gamma} \in \hat{\mathcal{B}}_{0,\eta}$ such that:

$$\hat{\Gamma}_i = \sum_{k=1}^{2r} \hat{u}_{ik} \hat{v}_k$$

with $\|u_i - \hat{u}_i\| \leq \frac{\eta}{2}$ and $\|v_k - \hat{v_k}\| \leq \frac{\eta}{2\sqrt{2r}}$ for every $i \in I$ and $k \in [2r]$.

In order to show that $\|\Gamma - \hat{\Gamma}\|_{1,2,\mathsf{T}} \leq \eta$, it is sufficient to show that $\|\hat{\Gamma}_i - \hat{\Gamma}_i\| \leq \eta$ for every $i \in [I]$.

$$
\begin{aligned}
\|\hat{\Gamma}_i - \Gamma_i\| &= \|\sum_{k=1}^{2r}(\hat{u}_{ik} - u_{ik})v_k + \sum_{k=1}^{2r} u_{ik}(v_k - \hat{v}_k)\| \\
&\leq \|\sum_{k=1}^{2r}(\hat{u}_{ik} - u_{ik})v_k\| + \|\sum_{k=1}^{2r} u_{ik}(v_k - \hat{v}_k)\| \\
&= \sqrt{\sum_{k=1}^{2r}(\hat{u}_{ik} - u_{ik})^2} + \|\sum_{k=1}^{2r} u_{ik}(v_k - \hat{v}_k)\| \leq \frac{\eta}{2} + \|\sum_{k=1}^{2r} u_{ik}(v_k - \hat{v}_k)\| \\
&\leq \frac{\eta}{2} + \sqrt{\sum_{k=1}^{2r} u_{ik}^2}\sqrt{\sum_{k=1}^{2r} \|v_k - \hat{v}_k\|^2} \leq \eta
\end{aligned}
\tag{58}
$$

Therefore $\hat{\mathcal{B}}_{0,\eta}$ is an $\eta$ net with respect to $\|\cdot\|_{1,2,\mathsf{T}}$. By Corollary 4.2.13 in Vershynin (2018), we can pick: $\left|\hat{\mathcal{S}}_{\frac{\eta}{2\sqrt{2r}}}(d)\right| \leq (\frac{4\sqrt{2r}}{\eta} + 1)^d$ and $|\hat{\mathcal{S}}_{\frac{\eta}{2}}(2r)| \leq (\frac{4}{\eta} + 1)^{2r}$ and conclude the bound on the cardinality of $\hat{\mathcal{B}}_{0,\eta}$.

$\square$

By Lemma 16 and a union bound,

$$
\begin{aligned}
\mathbb{P}\left(\inf_{\hat{\Gamma} \in \hat{\mathcal{B}}_{0,\eta}} L(\hat{\Gamma}, \Psi) < \frac{\zeta^4 \xi^2 |I|}{32Nd}\right) &\leq |\hat{\mathcal{B}}_{0,\eta}| \exp(-c\zeta^2\xi^2|I|K) \\
&\leq \exp\left(2dr\log(\frac{4\sqrt{2r}}{\eta} + 1) + 2|I|r\log(\frac{4}{\eta} + 1) - c\zeta^2\xi^2|I|K\right)
\end{aligned}
\tag{59}
$$

Therefore, whenever taking $|I| \geq \frac{N}{10}$ and $\eta = c_1\frac{\zeta^4\xi^2}{d}$ for some constant $c_1$ small enough, and combining equation 59 with equation 55, we conclude that whenever $K \geq \frac{C(r+\frac{dr}{N})}{\zeta^2\xi^2}\log\frac{d}{\zeta\xi}$ for a large enough constant $C$, we have:

$$
\mathbb{P}\left(\inf_{\Gamma \in \mathcal{B}_0(N,d,I,2r)} L(\Gamma, \Psi) < c_0\frac{\zeta^4\xi^2}{d}\right) \leq \exp(-c\zeta^2\xi^2 NK)
$$

$\square$

Now, consider $|I| \geq \frac{N}{10}$. The number of such sets $I$ is at-most $\exp(c_1 N)$ for some constant $c_1 > 0$. Therefore, applying Lemma 17 along with the union bound over all $I$ such that $|I| \geq \frac{N}{10}$ we have:

**Corollary 1.** *Under the conditions of Lemma 17, we have:*

$$
\inf_{\substack{I \subseteq N \\ |I| \geq \frac{N}{10}}} \inf_{\Gamma \in \mathcal{B}(N,d,I,2r)} L(\Gamma, \Psi) > c_0\frac{\zeta^4\xi^2}{d}
$$

*with probability at-least* $1 - \exp(-c\zeta^2\xi^2 NK)$

We are now ready to prove Lemma 13.

*Proof of Lemma 13.* Combining Lemma 15 and Corollary 1, we conclude that with probability at-least $1 - \exp(-c\zeta^2\xi^2 NK)$,

$$\mathcal{Y}(\Psi) \bigcap \left( \bigcup_{\substack{I \subseteq [N] \\ |I| \geq \frac{N}{10}}} \mathcal{M}(N, d, I, 2r) \right) = \emptyset$$

Note that if $\Delta \in \mathcal{Y}(\Psi)$ such that $|I_{\mathcal{Z}}(\Delta)| > \frac{N}{10}$ implies $\Delta \in \mathcal{M}(N, d, I, 2r)$ for some $|I| > \frac{N}{10}$. This allows us to conclude the statement of the lemma. $\qquad\square$

