# OpenReview forum: "Multi-User Reinforcement Learning with Low Rank Rewards"
_ICLR.cc/2023/Conference — Submitted to ICLR 2023_

### Official Review · Reviewer_r7Gm · 2022-10-19

**Confidence:** 3
**Correctness:** 3
**Technical Novelty And Significance:** 3
**Empirical Novelty And Significance:** Not applicable
**Recommendation:** 6

**Clarity, Quality, Novelty And Reproducibility:**

Clarity: The writing of the main paper and the appendix is poor according to the points above.

Quality: The proof of main theorems seems to have some problems and needs further clarification.

Novelty: This paper presents the novel and meaningful sample complexities improvements on the collaborative multi-user reinforcement learning problem.

Reproducibility: N.A.

**Strength And Weaknesses:**

Strength:

This paper designs novel efficient algorithms for collaborative multi-user reinforcement learning with low-rank reward matrices. The improvements in sample complexities from $O(N|S| |A|)$ to $O(N+|S| |A|)$ and from $O(Nd^{2})$ to $O(N+d)$ are very interesting and important for the research in this field.

Weaknesses:

Technical issues:
1. Since the authors adopt the big O notation like $O(N+|S| |A|)$ in the paper, I assume that this paper concerns the regime where $N$ and $|S| |A|$ both increase to infinity. The authors need to clarify whether they allow $r$ also increases according to Assumption 1. If so, then the exponential improvement claimed in Remark 1 is suspectable, since $r$ can also scale linearly in  $|S| |A|$. If not, such exponential improvement is unfair, since Dann and Brunskill 2015 does not impose such strong low-rank assumption. A minor typo is that $O(N+|S| |A|)$ is not fully correct, and it should be $\tilde{O}(N+|S| |A|)$ according to Theorem 1.

2. In the proof of Theorem 1, the value of $p$ needs more clarification. Lemmas 5 and 7 require the condition that $p<1/2$, which is not a trivial condition according to the presented proof. Concretely, in the proof of Theorem 1, it is required that $m\leq Np|\mathcal{G}_{h}^{c}/2|$, i.e., p\geq 2/(N|\mathcal{G}_{h}^{c}/2|) C_{1}\max(\mu_{1}^{2},\mu_{0})r(N+|\mathcal{G}_{h}^{c}/2|)\log^{2}|\mathcal{G}_{h}^{c}/2|\logH\delta. You need more assumptions on $r$ or $\mu_{1},\mu_{0}$ even to guarantee that $p\leq 1$. In such case, you can not let $C_{1}$ small enough, since it requires that $C_{1}$ is large enough in your proof.

3. For the proof of Theorem 3, could the author explain how to prove the first inequality in (48)? It seems not a trivial claim.


Writing issues:

1. The writing of the proof contains some vague words, and it is very hard to understand. For example, claim 3 needs the definition of the so-called 'correct marginal distribution'. I guess 'correct' means that they have the same distributions as original $J_{h}$ and $I_{h}$.

2. Please arrange the Appendix such that the statements of the proposition/lemma are before the proof of them. Otherwise, it is quite confusing with reading your deferred proofs part without the statement of what to prove.

3. The writing of the main paper is poor. There are a lot of typos, e.g. it should be 'smaller than the number of' in the first paragraph of the introduction. Also, please first state the equation/theorem and then refer to it, e.g. on the top of page 9, the reference to equation 2 is confusing.

I will be very happy to increase my scores if the authors resolve my technical concerns.

**Summary Of The Paper:**

This paper focuses on multi-user reinforcement learning with low-rank rewards. This paper considers the problem where all the users play in the MDPs with the same dynamics but different reward functions. Exploiting the low-rank property of the reward matrix, the authors design two novel sampling algorithms for tabular and linear MDPs respectively. Combined with the designed matrix completion algorithm with row-wise linear measurements, the sampling algorithms are shown to enjoy low sample complexities.

**Summary Of The Review:**

In summary, this paper presents interesting results on the collaborative multi-user reinforcement learning problem. Novel sampling algorithms are designed, and the improved sample complexities are derived. However, the presentation of the paper needs more improvements, and some technical issues remain to be solved.

---

> ### Author Response · Authors · 2022-11-13
> **Response to the Reviewer**
>
> We thank the reviewer for the technical comments and suggestions on overall presentation. We resolve the technical questions below and have incorporated the suggestions on the presentation in the revision. We are also working on making it better in terms of presentation. We hope the reviewer reconsiders their score as stated in the review.
>
> **Response to Technical comments:**
>
> 1)  As is common in collaborative filtering literature, we assume that the rank is constant and that $N$, $|S||A|$ are very large (we refere to low rank matrix factorization papers cited). Low rank matrix completion almost never deals with the regime $r \sim \min(N,|S||A|)$. We have clarified this aspect in our revised manuscript. The regime where $r$ scales linearly as $|S||A|$ is not relevant to our current work. But the non-asymptotic bounds involving $r$ still hold wherever relevant.
>
>
> 2) The most interesting scenario for collaborative filtering is when $p$ is very small. Therefore $p < 1/2$ is a very mild condition which is required for technical reasons only. It is a common assumption in collaborative filtering literature that the dimensions (N, |S||A|) are very large compared to the incoherence parameters ($\mu_1,\mu_2$) and that the rank $r = O(1)$. We seem to have missed specifying this in our work. We have added the condition that $|S||A|$ and $N$ are large enough (Theorem 1) in order to guarantee that $p < 1/2$.
>
> 3) The equation (48) follows from $\mathbb{E} f(S_h,A_h;x) \geq \tfrac{\zeta}{2}$ for every $x \in \mathcal{S}^{d-1}$. This fact has been proved in equation (47) and the function $f$ has been defined in Section 5. We have clarified this aspect in the revised manuscript.
>
> **Response to the Comments on Writing:**
>
> *“Please arrange the Appendix such that the statements of the proposition/lemma are before the proof of them. Otherwise, it is quite confusing with reading your deferred proofs part without the statement of what to prove.”*
>
> We chose this style of writing since proving technical lemmas by taking long detours can often obfuscate the main technical point we are trying to make.  We will try to make it as easy and logical as possible following some of the suggestions presented.

---

> > ### Comment · Reviewer_r7Gm · 2022-11-13
> > **Response to the Authors**
> >
> > Thanks a lot for your clarifications on the technical concerns.

---

### Official Review · Reviewer_iGHL · 2022-10-23

**Confidence:** 2
**Clarity, Quality, Novelty And Reproducibility:** 1.The paper is not easy to follow.

2…
**Correctness:** 3
**Technical Novelty And Significance:** 3
**Empirical Novelty And Significance:** 3
**Recommendation:** 5

**Strength And Weaknesses:**

Strength:

1.The motivation of this paper is very interesting. The low-rank assumption is very common in recommendation settings.

2.The theoretical result is sound and impressive.

Weakness:

1.The paper is not quite easy to follow. It is hard for people who are not familiar with reward free RL to get the main idea of the method.

2.The paper makes the low-rank assumption, but they do not justify whether it is reasonable.

3.The paper does not discuss it's limitation and conclude the paper.

4.It would be better to have a experiment to validate the effectiveness over baselines, even in a tabular game.

5.The collaborative MDP problem setting in the paper restrict the transition functions to be the same across MDPs, so it is equivalent to stating a single MDP with personalized reward function. The paper would have greater impact if it considers a more realistic setting where user transitions are non-deterministic and personalized as well.


Minor issues:
* Introduction, first paragraph, "is smaller the number" --> "is smaller than the number";
*
* Since you are borrowing the insight of collaborative filtering, in the motivating example, better briefly introduce the insight of your method. For example, stating that you can borrow the "exploration of similar users and assume that similar users generate similar responses."
* Problem setting, 2nd paragraph, "A_h = \pi_h(S_h)" --> "A_h \sim \pi_h(S_h)"
* Problem setting, "linear MDP setting" section, better state that \phi represents transition embedding and \psi represents reward embedding when first introduced.
* Section 3.2, Phase 2 step 1, "query obtain samples" --> "query/obtain samples"?
* An inconsistent notation: in problem setting you use e_i for different embedding dimensions, but in section 6 you state that i represents a user.


Questions to authors:
* Main points mentioned in weakness;
* Do you have explanation on why linear MDP has total complexity O(N+d)?
* Should it be G_h in the last two lines of Algorithm 1 instead of G?

**Summary Of The Paper:**

The paper considers the offline RL setting under N user-specific MDPs. The author addresses the sample efficiency and the exploration of collaborative reward function. The solution assumes a low-rank structure between user reward functions and use the idea of collaborative filtering to reduce the required number of samples for each user MDP.

A special problem definition:
* Same state/action space and transition function across users;
* Personalized reward function, assumed to be low-rank;
* Goal: learn optimal policies for N user-specific MDPs
* Goal2: improve sample efficiency.

The main intuition:
When multiple user MDPs are collaboratively learned, the low-rank nature can derive a solution that can collaborate the learnings of MDPs and reduce the number of sample required for each user MDP.

Proposed algorithm:
1. Reward free exploration: uniformly pick user MDP for a given trajectory and generate optimal policy function \Pi(R) and its estimator V(\Pi(R)).
2. Query the reward matrix: use these functions to sample and collect user-wise rewards of state-action pairs or sample linear measurements
3. Complete or estimate the reward matrix: recover the full matrix from the sampled records for each user;
4. Computing optimal policy: find the optimal policy on the full matrix.

Contribution:
* An RL algorithm with a proven sample efficiency boost.
* An algorithm for efficient matrix completion for tabular MDP.
* An algorithm for efficient policy sampling and estimation that satisfies given statistics.


**Summary Of The Review:**

Overall, the paper is interesting and novel. It is important to leverage the low-rank property of reward function in RL. But the clarity is low and there is no experimental result.

---

> ### Author Response · Authors · 2022-11-13
> **Response to the Reviewer**
>
> We thank the reviewer for the insightful comments. We have revised our manuscript along these lines. We have responded to specific comments and questions below. Since the reviewer states that the setting is interesting and that the results are impressive, we hope that they can reconsider their score.
>
> ---
>
> *“The paper is not quite easy to follow. It is hard for people who are not familiar with reward free RL to get the main idea of the method.”*
>
> We have added an explanation of why reward free RL is deployed in Section 3.
>
> ---
> *“The paper makes the low-rank assumption, but they do not justify whether it is reasonable.”*
> +
> *“It would be better to have a experiment to validate the effectiveness over baselines, even in a tabular game.”*
>
> We refer the reviewer to the common comments for a response. Here, we have answered this question and related questions by other reviewers.
>
> ---
> *“The collaborative MDP problem setting in the paper restricts the transition functions to be the same across MDPs, so it is equivalent to stating a single MDP with personalized reward function. The paper would have greater impact if it considers a more realistic setting where user transitions are non-deterministic and personalized as well.”*
>
> We want to clarify that the user transitions are not deterministic (since we allow random transitions). In case the reviewer is referring to the fact that a common transition matrix is shared across the users, we refer to the common response for a detailed explanation and additional references regarding this. Here, we refer to works which cluster users in practice and learn a common MDP for them. We want to stress that even the current setting is already technically sophisticated and gives us lots of insights into such settings. This also lays groundwork for further work which considers more complex models like the one the reviewer mentions.
>
> ---
>
> *"Do you have explanation on why linear MDP has total complexity O(N+d)?"*
>
> Samples are required to use
> (1) reward free exploration and
>
> (2) learn the reward function for each user.
>
> Since the users share a common MDP, (1) has a sample complexity of $O(d^2)$, which is a lower order term whenever N is large. Under low rank assumptions, the reward matrix has $O(r(N+d))$ free parameters which need to be estimated, and this leads to the said sample complexity.
>
> ---
>
> *“Should it be G_h in the last two lines of Algorithm 1 instead of G?”*
>
> Here, $G$ is indeed correct. We noticed that there was some confusion with the notation. Here, $G$ denotes the collection of all $(G_h)_{h\in [H]}$. We have clarified this in the revision.

---

> > ### Comment · Reviewer_iGHL · 2022-12-07
> > **Response to the Authors**
> >
> > Thanks for your detailed response.

---

> > > ### Author Response · Authors · 2022-12-09
> > > **Thank you**
> > >
> > > Thank you. We hope we have clarified all your concerns.

---

### Official Review · Reviewer_d781 · 2022-10-23

**Confidence:** 4
**Correctness:** 3
**Technical Novelty And Significance:** 3
**Empirical Novelty And Significance:** Not applicable
**Recommendation:** 6

**Clarity, Quality, Novelty And Reproducibility:**

My main concern with this work is its clarity. I believe some additional efforts can largely benefit the interpretation of the algorithms and results. Other than the presentation, I believe this work is of good quality and makes a reasonable contribution to the theoretical understanding of multi-agent RL. I have not checked all the proofs but from what I read, the authors involve sufficient proof details.

**Strength And Weaknesses:**

Strength:
+ The consideration of low-rank structure is reasonable and I believe is an important direction to be explored in RL, especially to break the curse of multi-agent;
+ This work nicely adopts existing techniques in reward-free explorations and matrix estimations, while also bringing new developments, especially in sampling strategies to guarantee specific sample properties and an extended matrix completion technique. The combination is inspiring;
+ I am not very familiar with matrix completion, but the proposed techniques (i.e., how to maintain the isotropic property in samples and how to perform matrix estimation with row-wise linear measurements) may be of independent merits.

Weakness:
I am overall satisfied with the technical side of this work. However, I found the presentation not very clear. One potential reason may be that I am more from an RL background instead of matrices analysis. However, I believe it is still important to illustrate key intuitions clearly. Especially, many interesting designs are left unexplained (some even not mentioned explicitly).
- For example, in the tabular setting, Algorithm 2 is designed to stop when rewards of important states are sufficiently collected, while the importance of states is measured by (in some sense) their "achievability" characterized by the value obtained from reward-free outputs. I believe this part is important; however, it is not mentioned in Sec. 3.1, instead in the analysis (i.e., beneath Assumption (Tab) 2), a connection between incoherence and the redundant states is discussed.
- Similarly, for linear MDP, no verbal explanation other than the algorithm box (Alg. 2) illustrates how to perform Step 1 in Phase 2, which essentially designs rewards in unexplored directions for the reward-free outputs to explore more.
- Also, the theoretical analysis part lacks certain illustrations. The two parts of Theorem 1 comes from reward-free exploration and reward query, respectively, while the choice of parameter p is also not intuitive (which seems to directly contribute to the second term). A similar illustration is also missing for linear MDP, as it is not explicitly mentioned the meaning of the three parts in the sample complexity.

Other than the above concerns on clarity, I also have certain confusion about the intuition and statement of the results. Hopefully, the authors can help me clarify these questions:
- Typically in RL studies, the main difficulty is to learn the transitions (thus many works even assume the rewards are known). However, in this work, the difficulty is in learning the rewards, as there are N agents with N different reward dynamics. Thus, for the results, the authors emphasize the regime of large N, which makes the cost for reward learning dominate. Then, the benefit seems to come from the shared transitions, as it does not depend on the identity of the agent. I am just wondering whether this is suitable/fair to compare with letting each agent learn individually and whether this is any more suitable baseline.
- It is a bit hard for me to interpret the results in Theorems 1 and 2. Especially, in Theorem 1, when the first term has a stronger dependency on epsilon; however the authors emphasize the dependency on the second term. For example, when epsilon = 1/N, the first term is dominating. For Theorem 2, it is even more complicated with more parameters involved. Hopefully, the authors can provide a more formal and detailed discussion of the results.
- I believe there should be works in matrix analysis handling noisy observations, and thus I am wondering whether this work can accommodate noisy rewards. If not, hopefully, the authors can illustrate the difficulties.

**Summary Of The Paper:**

This paper studies the problem of multi-agent reinforcement learning under the structure of low-rank rewards. Specifically, multiple agents share the same transition distributions while having different rewards, which are assumed to be represented by a low-rank matrix. Both tabular and linear MDPs are investigated.

The main idea is first to use reward-free algorithms to estimate transition structure (which is the same across different agents); then input designed rewards to the reward-free output to obtain exploration policies that only interest in collecting rewards (which are constructed to have reward samples satisfying specific properties); at last, use low-rank matrix estimation techniques to obtain the reward matrices for all agents from the samples in the second step.

The main technical difficulties are in how to obtain proper reward samples and how to use these samples. For tabular MDP, the idea is to encourage explorations of important states (determined by the reward-free outputs) and use existing techniques to perform the matrix estimation. The design in linear MDP is more challenging (thus more complicated) where the authors propose a newly crafted matrix estimation algorithm and carefully design the sampling strategy in MDP to meet the required sample properties.

With the above techniques, sample complexity analyses are performed on both tabular and linear MDP settings. Compared with learning individually, the designs are shown to be effective in leveraging the low-rank structure to decrease the sample complexity.

**Summary Of The Review:**

This work studies the problem of multi-agent RL, especially under the structure of low-rank rewards. Techniques are required from both RL and matrix analysis. On the RL side, for the shared transitions, reward-free explorations are first adopted, and later sample strategies are carefully constructed to obtain samples with certain properties. On the matrix analysis side, sampling requirements are first identified for the RL part and with these properties, existing techniques are adopted for the tabular setting while a new estimation technique is proposed for the linear setting. The combination of these techniques leads to performance improvements which are demonstrated theoretically. However, the presentation does not make me satisfied and I believe also does not fully demonstrate the techniques/results in this work as many key illustrations are missing, which leads to my recommendation.

---

> ### Author Response · Authors · 2022-11-13
> **Response to the Reviewer**
>
> We thank the reviewer for the thoughtful comments on the technical presentation. We have made improvements along these lines in the revised manuscript, by adding specific explanations at key places.
>
> *“The two parts of Theorem 1 comes from reward-free exploration and reward query, respectively, while the choice of parameter p is also not intuitive (which seems to directly contribute to the second term). A similar illustration is also missing for linear MDP, as it is not explicitly mentioned the meaning of the three parts in the sample complexity.”*
>
> The probability $p$ is chosen such that $p|\mathcal{S}||\mathcal{A}|N = \tilde{O}(r(|\mathcal{S}||\mathcal{A}|+N)) $, which is the number of free parameters required to describe a rank $r$ matrix. By a similar reasoning, the sample complexity for the linear MDP scales as $O(r(N+d))$.
>
> ---
> *“Typically in RL studies, the main difficulty is to learn the transitions (thus many works even assume the rewards are known). However, in this work, the difficulty is in learning the rewards, as there are N agents with N different reward dynamics. Thus, for the results, the authors emphasize the regime of large N, which makes the cost for reward learning dominate. Then, the benefit seems to come from the shared transitions, as it does not depend on the identity of the agent. I am just wondering whether this is suitable/fair to compare with letting each agent learn individually and whether this is any more suitable baseline.”*
>
> We refer the reviewer to the common response, where we have responded to the question of applicability of the problem setting along with responses to related questions from other reviewers. We have also added a detailed explanation in the revised manuscript.
>
> ---
>
> “It is a bit hard for me to interpret the results in Theorems 1 and 2. Especially, in Theorem 1, when the first term has a stronger dependency on epsilon; however the authors emphasize the dependency on the second term. For example, when epsilon = 1/N, the first term is dominating. For Theorem 2, it is even more complicated with more parameters involved. Hopefully, the authors can provide a more formal and detailed discussion of the results.”
>
> In this work, we are concerned with a very large $N$ (say a few millions) and a reasonable epsilon (say 0.1) in order to obtain a decent stateful policy. In a lot of MDPs, the optimal policy (or a reasonably good) can be obtained for finite $\epsilon$ instead of $\epsilon \to 0$, especially when there is a gap. This is the reason we emphasize the first term.
>
>  Either way, we note that individually learning each MDP separately costs $N|S||A|/epsilon^2$ without any low rank assumptions, which is worse than either of the terms given in the theoretical guarantees.
>
> ---
>
> *“I believe there should be works in matrix analysis handling noisy observations, and thus I am wondering whether this work can accommodate noisy rewards. If not, hopefully, the authors can illustrate the difficulties.”*
>
> Noisy matrix completion is less understood compared to the noiseless setting. The results for this case did not seem compatible with our RL setting. In the matrix completion setting (for Tabular RL) the recent work [5] requires the noise in each entry to be small to facilitate recovery (see Equation (7b) in [5]). While [7] considers recovery in the Frobenius norm which is not useful for the RL setting where a sup norm recovery is required.
>
> For the matrix estimation setting (like in Linear MDP), the standard results in [6] require restricted strong convexity (RSC). Finding RSC satisfying random measurements in the linear MDP does not seem to be easy. This is why we use a much weaker condition than RSC. We also note that RSC type assumptions cannot hold in our case since we have row wise dense measurements (i.e, a combination of sparse + dense). We believe that our current algorithm can handle noise (perhaps with a more specialized analysis or modifications). But we did not consider this setting for the sake of simplicity.
>
>
> [5] Noisy matrix completion: Understanding statistical guarantees for convex relaxation via nonconvex optimization, Chen et al
>
> [6] Estimation of (near) low-rank matrices with noise and high-dimensional scaling, Negahban and Wainwright
>
> [7] Noisy low-rank matrix completion with general sampling distribution, Olga Klopp
>
> ---

---

> > ### Comment · Reviewer_d781 · 2022-12-09
> > **Thank you for the response**
> >
> > I appreciate the authors' detailed and helpful feedback. After reading the feedback and the revised manuscript, my opinion remains unchanged that this work considers an important problem and provides insightful ideas; however, the current presentation can be further improved to better demonstrate these contributions and results.
> >
> > First, I still believe the algorithms can be better illustrated with more explanations and intuitions. For example, why use reward-free exploration, and why query the reward matrix following the designed procedure? These are important components and worth more discussions. In addition, the current paper can be reconstructed to help readers' understanding. For example, the reward-free exploration is used for algorithm design but introduced in problem formulation; Assumption (Tab/Lin) 1 is not used until the analysis and can be deferred to Section 4.
> >
> > Second, regarding the comparison with learning individually, I am wondering from where the performance improvement actually comes. Especially, as in the authors' feedback to Reviewer iGHL, the problem contains two parts: (1) learn the shared transition kernels; (2) learn the low-rank rewards. As the authors stated, since the users share a common MDP (in terms of transitions), the first part leads to a lower-order term in terms of N; the second term is then dominating in terms of N as different users have different rewards. This leads to the question that whether the improvement comes from the shared transitions or the low-rank rewards. It would be nice to involve some discussions on this point as this is one major contribution emphasized in the paper. For example, it would be helpful to consider baseline algorithms incorporating shared transitions without using the property of low-rank rewards. Another case is that the MDPs are the same for all users in transitions and rewards (i.e., rank-1 rewards?). It would be interesting to discuss whether the current results degenerate to the optimal performance in this extreme case.
> >
> > In summary, I appreciate the efforts of the authors and agree that certain contributions are made, and I am looking forward to a better-constructed paper for these contributions. I will increase the score to 6, but I am still holding a borderline opinion on this work.

---

> > > ### Author Response · Authors · 2022-12-10
> > > **Thank you.**
> > >
> > > Thank you for your feedback and raising the score. We will work on making the presentation much better and bring out the important technical aspects of the paper in an easily accessible manner.
> > >
> > > Regarding the importance of shared MDP vs. low rank rewards, we believe that both these assumptions are helpful in obtaining the improved sample complexity. This can be seen by counting the number of free parameters to be learned in order to describe the MDPs of all users.
> > >
> > > As noted in the common response, practitioners use clustering based on some known features like demographics in order to make sure that the MDP across the users inside a cluster remain the same. Low rank assumption is quite common for "one step rewards" in collaborative filtering. Both of these assumptions are made to reduce the dimensionality of the problem to ensure learning with sparse data.

---

### Official Review · Reviewer_HAK3 · 2022-10-31

**Confidence:** 3
**Correctness:** 3
**Technical Novelty And Significance:** 3
**Empirical Novelty And Significance:** Not applicable
**Recommendation:** 6

**Clarity, Quality, Novelty And Reproducibility:**


$\textbf{Clarity}$ & $\textbf{Quality}$: In general, it is joyful to follow the whole paper. The basic setups and contributions of the problem are well presented. The related literature about reinforcement learning and matrix completion is well-surveyed. However, insights into many assumptions and equations are not provided. This could make the readers difficult to understand some parts of the paper. Another minor comment is that the authors should cite some most recent works on low-rank matrix estimation.

$\textbf{Novelty}$: In general, the idea of this work is new and interesting. In particular, incorporating the low-rank structure into the study of reinforcement learning could significantly improve the sample complexity in learning. The only concern is whether it is reasonable to assume the low rankness of the reward matrix in reinforcement learning. The authors should provide more explanations on this.

$\textbf{Reproducibility}$: There are no experiments in this paper. The reproducibility could be significantly improved if the experiments are provided and the code could be released.

**Strength And Weaknesses:**

$\textbf{Strength}$: (1) In this work, the authors incorporated the low-rank structure into the algorithm design and sample complexity analysis of the multi-user reinforcement learning. It is interesting to see a significant improvement in sample complexity by fully exploiting the low-rank structure in the reward matrix. This could motivate researchers by incorporating other structures into reinforcement learning.

(2)  Generally, the paper is well-presented and easy to read. The main contributions are stated clearly.

$\textbf{Weaknesses}$: (1) There are many low-rank examples in collaborative filtering literature. However, it is rare to see examples of the low-rank reward matrix in reinforcement learning. It would be more convincing if the authors could provide some examples with explanations on the assumption that the reward matrix is low-rank in reinforcement learning. This is very important in clarifying the significance of this work.

(2)  The notation is a little confusing in this paper. For example, what is the definition of $\Delta$ in $\Delta(A)$?  What is $da$ in $\pi_h(da|s)$ on Page 3?

(3) Some assumptions in the paper are not well presented and motivated. For example, Are there any interpretations of Assumption (Lin) 4?

(4) It would be good if the authors could provide some instances of random vectors that satisfy the requirement in Eq. (2).



**Summary Of The Paper:**

In this work, the authors studied the problem of multi-user collaborative reinforcement learning under the assumption that the reward matrix is low-rank. They proposed a sample-efficient algorithm for tabular MDPs and linear MDPs by exploiting the low-rank structure to incorporate the techniques from matrix completion. In particular, they showed that the total sample complexity for every user scales as $O(N+|\mathcal{S}||\mathcal{A}|)$ instead of  $O(N|\mathcal{S}||\mathcal{A}|)$ in tabular MDPs and $O(N+d)$ instead of $O(Nd^2)$ in linear MDPs. This is a significant improvement.

**Summary Of The Review:**


At first, I have to claim that I am not an expert on reinforcement learning and I don’t know the related latest results. According to my understanding of this paper and the cited papers, the authors applied the techniques from matrix completion to the multi-user reinforcement learning problem. The theoretical contributions of this work are rather solid and strong. However, the discussions of the assumptions and theoretical results compared to the existing results are missing. Then, the readers don’t well understand the significance of these results.



$\textbf{Typo}$: (1) In Algorithm 1, “compete” should be “complete”.
(2) In Theorem 2, “Assumption” should be “Assumptions”.

---

> ### Author Response · Authors · 2022-11-13
> **Response to the Reviewer**
>
> We thank the reviewer for the helpful comments and insightful technical questions.
>
>  *“There are many low-rank examples in collaborative filtering literature. However, it is rare to see examples of the low-rank reward matrix in reinforcement learning. It would be more convincing if the authors could provide some examples with explanations on the assumption that the reward matrix is low-rank in reinforcement learning. This is very important in clarifying the significance of this work.”*
>
> We thank the reviewer for the feedback. We have added more discussion regarding the setting in our revised manuscript. We first refer the reviewer to the common response under the title “Applicability and Empirical Evaluation” - where we have elaborated on these points and other similar concerns raised by other reviewers.
>
> ---
>  *“The notation is a little confusing in this paper. For example, what is the definition of $\Delta$ in $\Delta(A)$? What is $da$ in $\pi_h(da|s)$ on Page 3?”*
>
> We understand that this can be confusing. We have clarified these aspects in the revision. Here, $\Delta(A)$ refers to the space of probability distribution over set $A$. $\pi_h(da|s)$ gives the conditional measure (or `density’) of action $a$ given the state $s$, which is the standard notation while considering the disintegration theorem for probability measures, which we realized can be confusing in this context.
>
>
> ---
> *“Some assumptions in the paper are not well presented and motivated. For example, Are there any interpretations of Assumption (Lin) 4?”*
>
> We refer to Appendix A, where we have dedicated an entire section describing and justifying this assumption. To summarize, we first demonstrate that the task of collaborative exploration requires a randomized policy. The usual policy class considered in the literature for reward maximization are deterministic. However, the space of all policies given by kernels $\pi_h(da|s)$ can be very large, and intractable when sets $S$ and $A$ are large/infinite. This can make it statistically hard to find policies to do the trick. Therefore, this assumption states that searching over a reasonable policy space is enough (here, reasonable means a tractable log-covering number which can ensure efficient statistical recovery). We also construct such policy classes in Appendix A.
>
> ---
>
> *“It would be good if the authors could provide some instances of random vectors that satisfy the requirement in Eq. (2)”
>
> The requirement (2) is satisfied by :
>
> a) The uniform distribution over the sphere
>
> b) Gaussian distribution $\mathcal{N}(0,I/2d)$, conditioned on the event that the norm is smaller than $1$
>
> c) Normalized Rademacher random vector
>
>  We note that this is a much less strict assumption than (a),(b) or (c) . This is also less stringent than the restricted strong convexity assumption which is usually used in sparse/low rank estimation [4] (our setting is also different in that we have row-wise dense measurements). To give some intuition, this means that given any vector $x$, there is some overlap between the random vector $\psi$ and $x$, ensuring that every measurement gives us some information helping us to complete the matrix. The third assumption is a standard bound on the covariance matrix.
>
> [4] Estimation of (near) low-rank matrices with noise and high-dimensional scaling. Negahban and Wainwright.
>
> ---

---

> > ### Comment · Reviewer_HAK3 · 2022-12-06
> > **Response to the Authors**
> >
> > Thanks for the authors' clarifications on the motivation and technical concerns.

---

> > > ### Author Response · Authors · 2022-12-09
> > > **Thank you.**
> > >
> > > Thank you. We hope we have clarified all your concerns.

---

### Official Review · Reviewer_XKWe · 2022-11-01

**Confidence:** 2
**Correctness:** 4
**Technical Novelty And Significance:** 3
**Empirical Novelty And Significance:** Not applicable
**Recommendation:** 6

**Clarity, Quality, Novelty And Reproducibility:**

The paper is well written. Breaking down the algorithm in different phases helps with parsing some of the key ideas/steps.


**Strength And Weaknesses:**

Strengths
1) The paper is generally well written.
1) The paper examines an interesting setting based on reasonable assumptions that may emerge in real applications.
2) The paper combines a lot of ideas from various lines of literature (e.g., MDPs, low rank matrix completion, multi-agent mean-field RL).


Weaknesses
1) The results do not seem particularly surprising given the prior work.
2) Although the authors try to incentivise the model via real world applications no effort is put into testing these ideas on an actual benchmark.





**Summary Of The Paper:**

The paper focus on collaborative multi-agent RL. In this setting, all agents share the same state-action space and transition probabilities but may have with different rewards. In this setting, they introduce an assumption inspired from collaborative filtering, that the reward matrix of the N users have low rank. Given this assumption, they provide algorithms for efficient learning in two settings: tabular MDPs
and linear MDPs. When the number of agents N is large and the rank is constant, the sample complexity per MDP depends logarithmically over the size of the state-space, which is exponentially better than the naive approach.

**Summary Of The Review:**

An interesting theoretical paper on multi-agent RL. The practical applications of the proposed scheme are less clear.

---

> ### Author Response · Authors · 2022-11-13
> **Response to the Reviewer**
>
> We thank the reviewer for the helpful comments on our work. We have responded to the two main concerns below.
>
>
> *“The results do not seem particularly surprising given the prior work.”*
>
> It would be helpful if the reviewer can elaborate on this comment, by pointing to some of such prior work. On a technical level, we believe that collaborative exploration methods for both the tabular setting and the linear MDP setting are novel. The connection to functional reward maximization and multi-agent mean-field RL for linear MDP motivates a lot of interesting future work, which we are pursuing at the moment.
>
> *“Although the authors try to incentivise the model via real world applications no effort is put into testing these ideas on an actual benchmark”*
>
>  We refer to the common response “Regarding Applicability and Empirical Evaluation”, where this concern and other similar concerns have been addressed.

---

> > ### Comment · Reviewer_XKWe · 2022-12-07
> > **Response**
> >
> > First of all, I do wish to ponit out that despite not being an expert in the area as my score shows I am on the positive side about the paper. The related work that I am referring is work coming from collaborative filtering where it seems some of the fundamentals ideas of the framework come. That being said, it is clear that some work is needed to adapt these ideas but at least from my admitedly non-expert perspective it was not clear to me what conceptual breakthroughs were needed along the way. That being said again, this is not a strong criticism on the paper but maybe a pointer to why I feel somewhat less excited about the work. As there more experts reviewers on this paper, I point to them for a conclusive evaluation of the technical novelty of the work.
> >
> > In regards to the applicability of the results, although I do understand that this setting does indeed have interestibng applications as my review explictily states "no effort is put into testing these ideas on an actual benchmark." I.e. there is not a single experiment even in a toy setting. I believe this is a clear drawback of the current work.
> >
> > Overall, I believe that my score remains unchanged. I.e. I am positive about the work but not in strong support.

---

> > > ### Author Response · Authors · 2022-12-09
> > > **Thank you.**
> > >
> > > We thank you for the feedback.
> > >
> > > Regarding the experiments on benchmarks: our work was mainly theoretical and serves to introduce this particular setting which combines collaborative filtering and RL. We want to explore testing on benchmarks in future work since a lot of the component parts involved (like reward free RL) have only been established theoretically and have not yet been implemented and explored in practice. This, we believe, will involve modifications and challenges which merit further research and a significant amount of effort.

---

### Author Response · Authors · 2022-11-13
**Response to the Reviews**

We thank the reviewers for their feedback and questions, which have helped us improve our manuscript. The most common concern from the reviewers was about the motivation behind the problem setting. We have elaborated more on this below and have modified the manuscript to add this discussion. We have replied to the specific comments by the reviewer in the respective replies.

**Regarding Setting, Applications, and Empirical Evaluation:**
We have added more discussion regarding the setting and its utility in our revised manuscript. Online learning and decision making with low-rank structure has gained  attention recently (see [1], [2] below). Here multiple bandit instances are collaboratively explored under low rank assumptions on their reward matrices.

In this work, we extend this setting to consider stateful modeling of such systems. In the context of e-commerce, this can allow the algorithm to discover temporal patterns like ‘User bought a Phone and hence they might be eventually interested in phone cover’ or ‘User last bought shoes many months ago which might be worn out by now, therefore recommend shoes’. Note that the fact that a user has bought a shoe changes their preferences (and hence the reward function). While we assume that the users share the same transition matrix, this can be relaxed in practice by clustering users based on side information and modeling each cluster to have a common transition matrix. This approach has been successfully deployed in various multi-agent RL problems in practice, including in sensitive healthcare settings (see [3] and reference therein).


We believe that the setting itself is quite insightful for stateful modifications of otherwise stateless models encountered in contextual bandits, and our main contribution is to obtain insights on the kind of considerations which are necessary when dealing with such multi-agent systems where each individual agent cannot be estimated efficiently. In the future, we intend to these observations in order to arrive at algorithms which work well with neural function approximation.


[1] Regret bounds and regimes of optimality for user-user and item-item collaborative filtering, Besler and Karzand

[2] Online Low Rank Matrix Completion, Pal and Jain

[3] Field Study in Deploying Restless Multi-Armed Bandits: Assisting Non-Profits in Improving Maternal and Child Health - Mate et al

---

### Author Response · Authors · 2022-11-18
**Regarding our Response**

We thank the reviewers again for their feedback regarding our work. We are happy that the reviewers find the technical contributions sound and novel. As the revision period is coming to a close, we were hoping to hear back from the reviewers regarding our response. We list some of the salient features of our work:

a) We introduce the collaborative reinforcement learning setting, which extends the recently considered problem of collaborative exploration in the bandit setting. This allows stateful modeling as described in our work.

b) We provide near-optimal algorithms in both tabular MDP and linear MDP settings, where the users explore collaboratively in order to learn the optimal policy for each user.

c) We introduce a novel low-rank matrix estimation algorithm which works in the semi-sparse regime encountered in the linear MDP setting.

We believe that the setting and the techniques which have been introduced are helpful in learning certain classes of multi-agent systems with temporal dynamics. Thank you.

---

### Author Response · Authors · 2022-12-01
**Gentle Reminder**

Since the discussion period is coming to a close, we were hoping to have a discussion with the reviewers regarding any concerns or observations they might have. It would be very helpful if the reviewers can respond over the next few days.

Best,
The Authors.

---

### Decision · Program_Chairs · 2023-01-20

**Decision:**

Reject

**Justification For Why Not Higher Score:**

There are two weaknesses that the reviewers and I feel should be fixed before the paper is ready for acceptance.

**Justification For Why Not Lower Score:**

Can't go lower

**Metareview: Summary, Strengths And Weaknesses:**

This paper consider a collaborative multi-agent RL problem where the reward matrix of the N users has a low-rank structure. The authors design algorithms that explore rewards collaboratively with N user-specific MDPs and can learn rewards efficiently in two key settings: tabular MDPs and linear MDPs. The sample complexity is reduced over non-collaborative algorithms. The reviewers and I found that the paper is well motivated and the results are generally sound.

This is a borderline paper that warranted a discussion among the reviewers, which I conducted. The discussion coalesced around the weaknesses of the paper which everyone felt should be addressed in a proper revision before it is published in a future conference.

1) Lack of experimental results even on toy datasets. The authors try to motivate the model via real world applications no effort is put into testing these ideas on an actual benchmark.

2) Lack of clarity in writing. Admittedly, this paper contains some complicated ideas, having to amalgamate terminologies from the RL literature and the matrix completion literature. However, several reviewers complained about the clarity of the writing and the vagueness in some of the mathematical exposition. They provided some concrete suggestions for the authors to improve their work on. Hence, I strongly suggest to the authors to take these into account for a future conference submission.

**Summary Of Ac-Reviewer Meeting:**

I conducted an email discussion with some of the reviewers. The reviewers agreed on the two weaknesses I highlighted above.